# Low-Rank Correction for Quantized LLMs

## Abstract

We consider the problem of model compression for Large Language Models (LLMs) at post-training time, where the task is to compress a well-trained model using only a small set of calibration input data. In this work, we introduce a new low-rank approach to correct for quantization errors of *activations* in LLMs: we propose to add low-rank weight matrices in full precision that act on the *unquantized* activations. We then solve a joint optimization problem over the quantized representation of the weights and additional low-rank weight matrices to quantize both weights and activations. We focus on the case of 4-bit weight-and-activation quantization (W4A4). Using ranks equivalent to 10% of the original weight matrix size, our approach reduces the accuracy gap with the original model by more than 50%. Using ranks equivalent to 30% of the original weight matrix, the accuracy gap is closed completely. We demonstrate our results on four recent LLMs, namely Llama-2, Llama-3, Phi-3 and Mixtral models.

## 1 Introduction

Large Language Models (LLMs) (Abdin et al., 2024; Dubey et al., 2024; Jiang et al., 2024) have demonstrated exceptional performances across a wide range of applications. However, due to their massive size, these models require considerable computational and memory resources at inference.

Post-training quantization (PTQ) is among the most important techniques to solve both memory and compute issues during LLM inference. The majority of quantization schemes focus on compressing LLMs by using weight-only quantization (Frantar et al., 2022; Shao et al., 2023; Dettmers et al., 2023b). One major limitation of PTQ is the presence of magnitude outliers in the model layer weights, which can severely affects the quantization process (Wei et al., 2022; Xiao et al., 2023) and deteriorate the performances of quantized models. To handle them, several offline approaches has been proposed in the literature, such as mixed-precision strategies (Dettmers et al., 2023b), adapted rescaling Lin et al. (2024), and incoherence processing Chee et al. (2024); Tseng et al. (2024). Recently, several works have introduced the use of supplementary low-rank weight matrices in full precision to mitigate quantization errors in weights Kang et al. (2024); Ou et al. (2024). This approach is analogous to the utilization of additional low-rank weight matrices for fine-tuning (Dettmers et al., 2023a; Hu et al., 2021).

Weight only quantization methods enable to store LLMs on smaller devices, and accelerate the General Matrix-Vector Multiply (GEMV) operators in the decoding stage Lin et al. (2024); Frantar et al. (2022), however, these approaches still require to keep activations in full precision (usually FP16). To improve on this, several works (Ashkboos et al., 2023; Xiao et al., 2023; Dettmers et al., 2022; Zhao et al., 2024) aim at quantizing both the weights and activations (and sometime KV cache) to compute the forward pass in low bit precision. Unlike weight quantization, the quantization of activations requires online strategies to compute their low bit representations on the fly (Jacob et al., 2018).To deal with outliers in activations, (Xiao et al., 2023) propose to scale the weights, thus reducing the magnitude range of activations. (Ashkboos et al., 2024; Liu et al., 2024) propose to process weights and activations using Hadamard transform. More recently Zhang et al. (2024) propose to add low-rank correction terms in full precision in order to correct for quantization errors. Although these approaches has been shown to be effective at W4A8, they still struggle to deal with the case where activations are quantized in lower bit precision such as W4A4, leaving an opportunity to improve on current methods in these harder settings.

**Contributions.** In this work, we improve on current SoTA approaches for PTQ, and introduce LRC, a new method that leverages low-rank weight matrices in full precision to correct for activation quantization errors. By jointly optimizing for quantized representations of the original weights and full precision low-rank weights matrices correcting the errors of activation quantization, our method allows for quantizing weights, and activations (and KV caches) to 4 bits with minimal loss in accuracy. Our main contributions are summarized below.

- We introduce a general framework that aims at jointly optimizing for quantized representations of the original weights acting on the quantized activations, as well as full precision low-rank weights matrices operating on the *unquantized* activations.

- We derive an alternative scheme to solve the joint optimization problem and obtain a simple algorithm easily compatible with recent quantization techniques, such as QuaRot (Ashkboos et al., 2024) and GPTQ (Frantar et al., 2022).

- Using our quantization scheme, LRC manages to reduce the accuracy gap with original models by more than 50% using only low-rank matrices with $10\%$ of the original size, and outperform all current approaches at W4A4.

## 1.1 RELATED WORK

**Dealing with Outliers.** One major limitation of quantization techniques is the presence of outliers in both weights and activations that can deteriorate the quality of the approximations, and lead to a considerable drop in performance. A recent line of works (Ashkboos et al., 2024; Liu et al., 2024) proposed to apply specific rotations (and their inverses), on both weights and activations in order to remove outliers while preserving exactly the output of the original model. Such a pre-processing step, in combination with the GPTQ Algorithm (Frantar et al., 2022) enabling efficient quantization of the weights, and lead to SoTA performances in quantization at W4A8. In this work, we leverage this pre-processing step, and consider these methods as baselines for our approach where low-rank weight matrices in full precisions are added to the forward pass in order to correct for quantization errors on activations, enabling to quantize LLMs at W4A4 with a 50% gain.

**Low-Rank Correction.** Recent works proposed to leverage low-rank matrices to reduce quantization errors in LLMs (Zhang et al., 2024; Ou et al., 2024; Saha et al., 2024). In (Zhang et al., 2024), the authors consider the case where both weights and activations are quantized, and propose to add well-chosen low-rank matrices in full precision in the forward pass to reduce the quantization errors. To deal with outliers, they first compute some rescaling matrices (Lin et al., 2024; Xiao et al., 2023) using a calibration dataset, and then propose to add low-rank approximation of the rescaled residual errors between the original and quantized weights using SVD. Although the rescaling process leverages statistical properties of the activations to remove outliers, Zhang et al. (2024) do not exploit them for computing their low-rank correction terms. In (Ou et al., 2024), the authors improve on the low-rank approach introduced in Zhang et al. (2024) and propose to apply a PCA on the output activation errors using a calibration dataset. In (Saha et al., 2024), the authors consider a joint formulation of the quantization problem to optimize for both the quantized weights and the low-rank terms. However, in these two works, the authors only focus on the quantization of the weights, leaving aside the quantization of activations. In this work, we also consider a joint formulation, however our main focus is on improving the quantization of activations. We improve on prior research by incorporating both the empirical distribution of activations and the errors induced by activation quantization into our analysis to optimize the low-rank weight matrices.

## 2 BACKGROUND ON POST-TRAINING QUANTIZATION

**Weight Quantization.** Weight quantization aims at obtaining new weights with a lower bit precision, reducing the memory needed to store the model, while preserving the output of the original model. One standard approach is to perform layer-wise quantization, where quantized weights are obtained by solving a reconstruction problem on a calibration dataset. Given a small dataset of $n$ sampled activations $\boldsymbol{X}_\ell \in \mathbb{R}^{d_\ell^{\mathrm{in}} \times n}$ at a certain layer $\ell$, and the associated weight matrix $\boldsymbol{W}_\ell \in \mathbb{R}^{d_\ell^{\mathrm{out}} \times d_\ell^{\mathrm{in}}}$, the goal is to find a matrix of quantized weights $\widehat{\boldsymbol{W}}_\ell$ which minimizes the quadratic

error over the dataset (Nagel et al., 2020):

$$\min_{\widehat{\boldsymbol{W}}_\ell \in \mathcal{C}(b) \cap \mathbb{R}^{d_\ell^{\text{out}} \times d_\ell^{\text{in}}}} \mathcal{L}_{\text{q}}(\widehat{\boldsymbol{W}}_\ell) := \|\boldsymbol{W}_\ell \boldsymbol{X}_\ell - \widehat{\boldsymbol{W}}_\ell \boldsymbol{X}_\ell\|_2^2 , \tag{1}$$

where $\mathcal{C}(b)$ is the constraint set of matrices admitting a certain bit per weight $b > 0$ precision. Due to the non-convexity of the constraints, finding the exact solution of the problem is hard, and many works have focused on designing algorithms to efficiently approximate a solution (Frantar et al., 2022; van Baalen et al., 2024; Lin et al., 2024; Egiazarian et al., 2024).

**Activation Quantization.**  While weight quantization enables to store large models at a lower memory cost, it still requires additional memory space at inference time to perform the forward operations in full precision, usually by de-quantizing weights to FP16. To reduce the memory footprint and FLOP cost further, it is desirable to quantize the activations during inference to compute matrix multiplications in lower precision, usually using specialized cuda kernels that perform GEMM in low bit precision (Wei et al., 2022; Yao et al., 2022; Wu et al., 2023; Xiao et al., 2023). The quantization step of activations generally happens at every layer of a model as one might want to preserve full precision activations to perform coordinate-wise non-linear transformations of the activations (Ashkboos et al., 2024; Xiao et al., 2023). Additionally, as quantizing activations require additional operations in the forward, this step must only incur a small overhead in time and memory. Due to this constraint, previous work consider simple, yet efficient techniques such as round-to-nearest (RTN) (Ashkboos et al., 2024) to quantize activations on the fly at inference time.

In our work we assume a simple on-the-fly quantization, rescaling each activation $\boldsymbol{x}$ by $c \cdot \max(\text{abs}(\boldsymbol{x}))$ and rounding to the nearest integer. We perform a simple hyper-parameter search for $c$. Our main focus is deriving a low-rank correction to the weight matrix that accounts for some of the error incurred in activation quantization.

## 3 LOW-RANK CORRECTION

In this work, we propose to add full precision low-rank weight matrices in the forward pass that act on the *unquantized* activations to correct for quantization errors of activations. In the following, we start by presenting the proposed framework to quantize a single weight matrix while correcting for quantization errors using additional low-rank weight matrices. Then we introduce our proposed algorithm that effectively computes the quantized representations of the original weights in low-bit precision and the low-rank weight matrices in full precision. We conclude this section by providing an overview of the approach when applied on a standard LLM.

### 3.1 GENERAL FRAMEWORK

Before introducing our framework, we need to establish some clarifying notations.

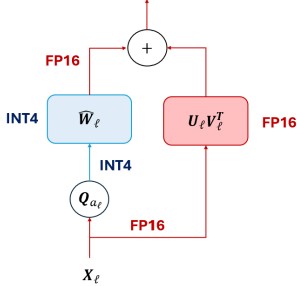

**Notations.**  Given a calibration dataset $\boldsymbol{X} \in \{1, ..., D\}^n$ of token ids where $D$ is the dictionary size, and a model $\mathcal{M}$ with $L$ layers, we denote $(\boldsymbol{X}_\ell)_{1 \leq \ell \leq L}$ where $\boldsymbol{X}_\ell \in \mathbb{R}^{d_\ell^{\text{in}} \times n}$, the sequence of activations obtained along the forward pass of $\mathcal{M}$ at each layer $\ell$ when applied on $\boldsymbol{X}$. We also denote $(\boldsymbol{W}_\ell)_{1 \leq \ell \leq L}$ where $\boldsymbol{W}_\ell \in \mathbb{R}^{d_\ell^{\text{out}} \times d_\ell^{\text{in}}}$, the sequence of weight matrices of $\mathcal{M}$ which act on the activations $(\boldsymbol{X}_\ell)_{1 \leq \ell \leq L}$ respectively along the forward pass. In the following, we always consider the case where $n \geq \max_{1 \leq \ell \leq L}(d_\ell^{\text{out}}, d_\ell^{\text{in}})$, as in

Figure 1: Computational scheme of our method, where both weights and activations are quantized, and a low-rank matrix in full precision is added and operates on the *unquantized* activations.

practice we use 128 sentences of 2048 tokens, giving $n \simeq 200k$ while in general $d \simeq 10k$. We also denote for $b > 0$, $\mathcal{C}(b)$ the constraint set of matrices admitting a certain bit per weight $b > 0$ precision, and $Q_b(\cdot)$ any quantization operator, that given an input matrix $\boldsymbol{Z} \in \mathbb{R}^{d \times d'}$ produces a matrix of same shape and satisfying $Q_b(\boldsymbol{Z}) \in \mathcal{C}(b)$. Finally, we call two optimization problems equivalent if they have the same solutions.

**Low-Rank Correction Problem.** We propose to extend the framework introduced in equation 1 for layer-wise quantization of weight matrices only, by also accounting for the errors induced by the quantization of activations and by adding low-rank weight matrices to correct them. Given the original activations $\boldsymbol{X}_\ell$ obtained at layer $\ell$, the associated weight matrix $\boldsymbol{W}_\ell$, a bit precision for the weights $b_\ell > 0$, a bit precision for the activations $a_\ell > 0$, a quantization operator $Q_{a_\ell}(\cdot)$ acting on the activations at inference time, and a rank $1 \le k_\ell \le \min(d_\ell^{\text{in}}, d_\ell^{\text{out}})$, we propose to consider the following reconstruction problem:

$$\min_{\substack{\widehat{\boldsymbol{W}}_\ell \in \mathcal{C}(b_\ell) \cap \mathbb{R}^{d_\ell^{\text{out}} \times d_\ell^{\text{in}}}, \\ \boldsymbol{U}_\ell \in \mathbb{R}^{d_\ell^{\text{out}} \times k_\ell}, \boldsymbol{V}_\ell \in \mathbb{R}^{d_\ell^{\text{in}} \times k_\ell}}} \mathcal{L}_{\text{qlr}}(\widehat{\boldsymbol{W}}_\ell, \boldsymbol{U}_\ell, \boldsymbol{V}_\ell) := \|\boldsymbol{W}_\ell \boldsymbol{X}_\ell - \widehat{\boldsymbol{W}}_\ell Q_{a_\ell}(\boldsymbol{X}_\ell) - \boldsymbol{U}_\ell \boldsymbol{V}_\ell^\top \boldsymbol{X}_\ell\|_2^2 . \quad (2)$$

Our goal here is to jointly optimize for both a quantized weight matrix $\widehat{\boldsymbol{W}}_\ell$ acting on the quantized activations and full precision low-rank weight matrices $\boldsymbol{U}_\ell, \boldsymbol{V}_\ell$ acting on the unquantized activations in order to reconstruct the original output of the weight matrix $\boldsymbol{W}_\ell$.

**Comparison between $\mathcal{L}_{\text{qlr}}$ and $\mathcal{L}_{\text{q}}$.** While similar in nature with the optimization problem introduced in equation 1, the reconstruction problem defined in equation 2 presents two major differences: (i) we propose to account for the quantization errors of activations by considering the quantized activation $Q_{a_\ell}(\boldsymbol{X}_\ell)$ as input to the quantized weight $\widehat{\boldsymbol{W}}_\ell$, rather than the original activation $\boldsymbol{X}_\ell$. (ii) Additionally, we propose to correct the quantization errors of activations using a low-rank correction matrix $\boldsymbol{U}_\ell \boldsymbol{V}_\ell^\top$ in full precision and applied on the *unquantized* activations $\boldsymbol{X}_\ell$. Figure 1 illustrates the computational scheme proposed in this work.

## 3.2 LRC Algorithm

Let us now present the proposed algorithm to solve equation 2. In the following we drop the dependency on $\ell$ of our notations for better readability. Starting from initial low-rank weight matrices $\boldsymbol{U}^{(0)}, \boldsymbol{V}^{(0)}$, we propose to alternatively optimize for $\widehat{\boldsymbol{W}}$ and $(\boldsymbol{U}, \boldsymbol{V})$ according to $\mathcal{L}_{\text{qlr}}$. For $t = 1, \ldots, T$, we propose to perform the following updates:

$$\widehat{\boldsymbol{W}}^{(t)} := \arg\min_{\widehat{\boldsymbol{W}} \in \mathcal{C}(b) \cap \mathbb{R}^{d^{\text{out}} \times d^{\text{in}}}} \|\boldsymbol{W}\boldsymbol{X} - \widehat{\boldsymbol{W}} Q_a(\boldsymbol{X}) - \boldsymbol{U}^{(t-1)}(\boldsymbol{V}^{(t-1)})^\top \boldsymbol{X}\|_2^2 \quad (3)$$

$$\boldsymbol{U}^{(t)}, \boldsymbol{V}^{(t)} := \arg\min_{\boldsymbol{U} \in \mathbb{R}^{d^{\text{out}} \times k}, \boldsymbol{V} \in \mathbb{R}^{d^{\text{in}} \times k}} \|\boldsymbol{W}\boldsymbol{X} - \widehat{\boldsymbol{W}}^{(t)} Q_a(\boldsymbol{X}) - \boldsymbol{U}\boldsymbol{V}^\top \boldsymbol{X}\|_2^2 . \quad (4)$$

**On the Update of $\widehat{\boldsymbol{W}}$.** To update $\widehat{\boldsymbol{W}}$, we rely on already existing solvers addressing equation 1. We show in the next Proposition that the optimization problem defined in equation 3 can be equivalently reformulated as a standard layer-wise quantization problem as defined in equation 1.

**Proposition 3.1.** *Let us denote $\boldsymbol{Y} := Q_a(\boldsymbol{X}) \in \mathcal{C}(a) \cap \mathbb{R}^{d^{\text{in}} \times n}$, and assume $\boldsymbol{Y}$ is full rank. Then, by denoting $\widetilde{\boldsymbol{W}}^{(t)} := (\boldsymbol{W} - \boldsymbol{U}^{(t)}(\boldsymbol{V}^{(t)})^\top)\boldsymbol{X}\boldsymbol{Y}^\top(\boldsymbol{Y}\boldsymbol{Y}^\top)^{-1}$, we have that the optimization problem defined in equation 3 is equivalent to the following reconstruction problem:*

$$\min_{\widehat{\boldsymbol{W}} \in \mathcal{C}(b) \cap \mathbb{R}^{d^{\text{out}} \times d^{\text{in}}}} \|\widetilde{\boldsymbol{W}}^{(t)} \boldsymbol{Y} - \widehat{\boldsymbol{W}} \boldsymbol{Y}\|_2^2 . \quad (5)$$

Therefore, updating $\widehat{\boldsymbol{W}}$ according to equation 3, is equivalent to solving equation 5, which can be approximated by using any solvers designed for equation 1 such as (Frantar et al., 2022; Lin et al., 2024; Egiazarian et al., 2024). In practice, we use the GPTQ algorithm (Frantar et al., 2022) that only requires access to the target weight matrix $\widetilde{\boldsymbol{W}}^{(t)}$ and the covariance matrix $\boldsymbol{Y}\boldsymbol{Y}^\top$. Algorithm 2 presented in Appendix B summarizes this step.

*Remark* 3.2. It is important to highlight that the choice of GPTQ (Frantar et al., 2022) to solve of equation 5 is arbitrary. Any alternative solver capable of efficiently addressing the problem formulated in equation 5 could be employed in its place.

**On the Update of $\boldsymbol{U}, \boldsymbol{V}$.** While solving exactly equations 1, or 5 is still an open question due to the non-convexity of the constraints, obtaining the update for $\boldsymbol{U}, \boldsymbol{V}$, that is solving equation 4, can be done in closed form, as shown in the following Proposition.

**Proposition 3.3.** *Assume that $\boldsymbol{X}$ is full rank and let us denote $\boldsymbol{Y} := Q_a(\boldsymbol{X})$. Then the optimization problem defined in equation 4 is equivalent to the following optimization problem:*

$$\max_{\boldsymbol{U} \in \mathbb{R}^{d^{out} \times k} \cap \mathcal{O}, \boldsymbol{V} \in \mathbb{R}^{d^{in} \times k}} Tr(\boldsymbol{U}^\top (\boldsymbol{\Sigma_1} + \boldsymbol{\Sigma_2^{(t)}} - \boldsymbol{\Sigma_3^{(t)}})\boldsymbol{U})$$

$$s.t. \ \boldsymbol{V} = \left[ \boldsymbol{W}^\top - (\boldsymbol{X}\boldsymbol{X}^\top)^{-1}\boldsymbol{X}\boldsymbol{Y}^\top (\widehat{\boldsymbol{W}}^{(t)})^\top \right] \boldsymbol{U} \ , \quad (6)$$

*where $\mathcal{O}$ is the set of matrices with orthnormal columns,*

$$\boldsymbol{\Sigma_1} := \boldsymbol{W}\boldsymbol{X}\boldsymbol{X}^\top \boldsymbol{W}^\top, \ \ \boldsymbol{\Sigma_2^{(t)}} := \widehat{\boldsymbol{W}}^{(t)}\boldsymbol{Y}\boldsymbol{X}^\top (\boldsymbol{X}\boldsymbol{X}^\top)^{-1}\boldsymbol{X}\boldsymbol{Y}^\top (\widehat{\boldsymbol{W}}^{(t)})^\top, \ \ and$$

$$\boldsymbol{\Sigma_3^{(t)}} := \widehat{\boldsymbol{W}}^{(t)}\boldsymbol{Y}\boldsymbol{X}^\top \boldsymbol{W}^\top + \boldsymbol{W}\boldsymbol{X}\boldsymbol{Y}^\top (\widehat{\boldsymbol{W}}^{(t)})^\top \ .$$

*In addition, a solution can be obtained by defining $\boldsymbol{U}$ as the $k$ unit eigenvectors of $\boldsymbol{\Sigma}^{(t)} := \boldsymbol{\Sigma_1} + \boldsymbol{\Sigma_2^{(t)}} - \boldsymbol{\Sigma_3^{(t)}}$ associated to its $k$ largest eigenvalues, and $\boldsymbol{V}$ as in equation 6.*

To compute the updated $\boldsymbol{U}^{(t)}, \boldsymbol{V}^{(t)}$, we require an access to the original weight matrix $\boldsymbol{W}$, the current quantized approximation $\widehat{\boldsymbol{W}}^{(t)}$, and the covariance and cross-covariance $\boldsymbol{X}\boldsymbol{X}^\top$ and $\boldsymbol{X}\boldsymbol{Y}^\top$ respectively. The pseudo-code of this step is detailed in Algorithm 3 in Appendix B.

**Initialization.** To initialize our algorithm, that is to instantiate $\boldsymbol{U}^{(0)}$ and $\boldsymbol{V}^{(0)}$, we propose to consider a relaxed formulation of the original optimization problem defined in equation 2, where we remove the constraint on $\widehat{\boldsymbol{W}}$. More formally we consider the following optimization problem:

$$\min_{\substack{\widetilde{\boldsymbol{W}} \in \mathbb{R}^{d^{out} \times d^{in}}, \\ \boldsymbol{U} \in \mathbb{R}^{d^{out} \times k}, \boldsymbol{V} \in \mathbb{R}^{d^{in} \times k}}} \mathcal{L}_{\mathrm{qlr}}(\widetilde{\boldsymbol{W}}, \boldsymbol{U}, \boldsymbol{V}) \ . \quad (7)$$

In fact, equation 7 can be solved in closed form as shown in the following Proposition.

**Proposition 3.4.** *Let us denote $\boldsymbol{Y} := Q_a(\boldsymbol{X})$ and assume that $\boldsymbol{Y}$ is full rank. Then the optimization problem defined in equation 7 is equivalent to the following optimization problem:*

$$\max_{\substack{\widetilde{\boldsymbol{W}} \in \mathbb{R}^{d^{out} \times d^{in}}, \\ \boldsymbol{U} \in \mathbb{R}^{d^{out} \times k} \cap \mathcal{O}, \boldsymbol{V} \in \mathbb{R}^{d^{in} \times k}}} Tr(\boldsymbol{U}^\top \boldsymbol{\Sigma}_{init} \boldsymbol{U})$$

$$s.t. \ \boldsymbol{V} = \boldsymbol{W}^\top \boldsymbol{U} \ \ and \ \ \widetilde{\boldsymbol{W}} = \left[ \boldsymbol{W} - \boldsymbol{U}\boldsymbol{V}^\top \right] \boldsymbol{X}\boldsymbol{Y}^\top (\boldsymbol{Y}\boldsymbol{Y}^\top)^{-1} \ , \quad (8)$$

*where $\mathcal{O}$ is the set of matrices with orthnormal columns, and $\boldsymbol{\Sigma}_{init} := \boldsymbol{W}\boldsymbol{X}[I_n - \boldsymbol{Y}^\top (\boldsymbol{Y}\boldsymbol{Y}^\top)^{-1}\boldsymbol{Y}]\boldsymbol{X}^\top \boldsymbol{W}^\top$. In addition, a solution can be obtained by defining $\boldsymbol{U}$ as the $k$ unit eigenvectors of $\boldsymbol{\Sigma}_{init}$ corresponding to the $k$ largest eigenvalues, and $\boldsymbol{V}$ and $\widetilde{\boldsymbol{W}}$ as in equation 8.*

Therefore we can initialize $\boldsymbol{U}^{(0)}$ and $\boldsymbol{V}^{(0)}$ according to equation 7 in closed form. We present the initialization step in Algorithm 4 of Appendix B. It is worth noting that the solution $\widetilde{\boldsymbol{W}}$ obtained in equation 8 can be used to measure an oracle performance of our alternating minimization scheme, i.e. the effect of correcting for activation quantization, assuming a perfect weight quantizer.

*Remark* 3.5. Observe that the update on $\widehat{\boldsymbol{W}}$ obtained in equation 5 aims at quantizing $\widetilde{\boldsymbol{W}}^{(t)}$, which can be seen as the optimal unconstrained weight matrix when the low-rank correction matrices are fixed and set to $\boldsymbol{U}^{(t)}, \boldsymbol{V}^{(t)}$.

**Numerical Stability.** Let us now discuss the assumptions made in our previous results. In the proposed updates, we either need $\boldsymbol{X}\boldsymbol{X}^\top$, or $\boldsymbol{Y}\boldsymbol{Y}^\top$ to be full rank. To avoid the case where these matrices are singular, we add a regularization term to these matrices, and consider instead:

$$\boldsymbol{\Sigma}_x := \boldsymbol{X}\boldsymbol{X}^\top + \epsilon_x I_{d^{in}}, \ \ and \ \ \boldsymbol{\Sigma}_y := \boldsymbol{Y}\boldsymbol{Y}^\top + \epsilon_y I_{d^{in}} \ ,$$

where $I_d$ denotes the identity matrix of size $d$. In practice we set $\epsilon_x := \frac{1e-2}{d^{in}} Tr(\boldsymbol{X}\boldsymbol{X}^\top)$, and similarly, $\epsilon_y := \frac{1e-2}{d^{in}} Tr(\boldsymbol{Y}\boldsymbol{Y}^\top)$.

**LRC Algorithm.** Finally, we present the full algorithm for LRC 1 that aims at approximating a solution of equation 2. Note that in practice, we accumulate batches of activations $\boldsymbol{X}$ to avoid running out of memory, and update $\boldsymbol{\Sigma}_x, \boldsymbol{\Sigma}_y, \boldsymbol{\Sigma}_{xy}$, as defined in lines 3, 4, 5, in an online fashion before initializing $\boldsymbol{U}, \boldsymbol{V}$ in line 6.

---

**Algorithm 1** LRC($\boldsymbol{W}, \boldsymbol{X}, b, a, k$)

---

1: **Input:** Original weight matrix $\boldsymbol{W}$, activation $\boldsymbol{X}$, the bit precision for weights $b$, the bit precision for activation $a$, the rank $k$, and number of iterations $T$.
2: **Output:** Quantized weight matrix $\widehat{\boldsymbol{W}}$, low-rank weight matrices $\boldsymbol{U}, \boldsymbol{V}$
3: $\boldsymbol{\Sigma}_x \leftarrow \boldsymbol{X}\boldsymbol{X}^\top + \epsilon_x \mathrm{I}_{d^{\mathrm{in}}}$
4: $\boldsymbol{Y} \leftarrow Q_a(\boldsymbol{X}), \quad \boldsymbol{\Sigma}_y \leftarrow \boldsymbol{Y}\boldsymbol{Y}^\top + \epsilon_y \mathrm{I}_{d^{\mathrm{in}}}$
5: $\boldsymbol{\Sigma}_{xy} \leftarrow \boldsymbol{X}\boldsymbol{Y}^\top$
6: $\boldsymbol{U}, \boldsymbol{V} \leftarrow$ Init-LR($\boldsymbol{W}, \boldsymbol{\Sigma}_x, \boldsymbol{\Sigma}_y, \boldsymbol{\Sigma}_{xy}, k$),   using Alg. 4
7: **for** $t = 1, ..., T$ **do**
8:    $\widehat{\boldsymbol{W}} \leftarrow$ Update-Quant($\boldsymbol{W}, \boldsymbol{U}, \boldsymbol{V}, \boldsymbol{\Sigma}_y, \boldsymbol{\Sigma}_{xy}, b$),   using Alg. 2
9:    $\boldsymbol{U}, \boldsymbol{V} \leftarrow$ Update-LR($\boldsymbol{W}, \widehat{\boldsymbol{W}}, \boldsymbol{\Sigma}_x, \boldsymbol{\Sigma}_{xy}, k$),   using Alg. 3
10: **end for**
11: **return** $\widehat{\boldsymbol{W}}, \boldsymbol{U}, \boldsymbol{V}$

---

**Application of LRC on LLMs.** LRC consists of two stages: (1) the model is first pre-processed according to the QuaRot procedure (Ashkboos et al., 2024), where Hadamard rotation matrices are fused with the weights to reduce the incoherence of both weights and activations, while maintaining the outputs of the original model. (2) Then, the model is quantized using the LRC algorithm 1, where both weights and activations are quantized, and optimized low-rank matrices in FP16 precision are added to the forward pass of each weight matrix.

To compute the Hessians ($\boldsymbol{\Sigma}_{xy}, \boldsymbol{\Sigma}_x, \boldsymbol{\Sigma}_y$) we follow Frantar et al. (2022) in using calibration dataset taken from Wikitext-2: 128 randomly selected sequences of length 2048. We found that computation of these matrices required 64-bit precision for numerical accuracy. By default, our subroutine for quantization follows GPTQ (Frantar et al., 2022), (we study other subroutines in the following section). LRC works sequentially through the weight matrices of the model, computing activations for each weight matrix, obtaining the covariance and cross-covariances matrices needed to apply Algorithms 1 and solving the optimization problem 2 for each before moving to the next layer.

## 4 EXPERIMENTS

This section aims to achieve two primary objectives: (i) to demonstrate that LRC reduces the accuracy gap with the original models by more than 50% while utilizing low-rank matrices with only 10% of the original size, and surpasses all existing methods under W4A4 settings; and (ii) to show that when ranks corresponding to 30% of the original weight matrix are used in LRC, the accuracy gap is fully eliminated. Furthermore, we establish that when activations remain unquantized and only weights are quantized, the inclusion of low-rank correction terms becomes unnecessary.

In all our experiments, we build on top of the QuaRot (Ashkboos et al., 2024) codebase, extending the method to include our approach. All our experiments focus on 4-bit quantization. We experiment with quantization of Phi-3 (mini-4k-instruct), LLama-2 (7 and 13 B), Llama-3 (8B) and Mixtral (8x7B). All results in the table are simulated using Pytorch.

### 4.1 BENCHMARK

We first present our main results. Our ambition is to close the gap between our main benchmark, QuaRot, and the original FP16 model by adding ranks equivalent to 10% of the original weight matrix size. To show the effect of the LRC algorithm relative to previous approaches (Zhang et al., 2024; Ou et al., 2024) we also consider a baseline of the QuaRot approach with SVD applied to the weight-matrix error (we denote this approach as SVD in tables 1, 2 and 3).

We apply our LRC approach with a single iteration (denoted LRC (1)) and 5 iterations (LRC (5)). The runtime of our approach is comparable to the QuaRot method, though we require additional memory to store the statisitcs of the activations. Quantizing the Mixtral model on 4xA100 GPUs required 7 hours to complete 1 iteration, or 9 hours to complete 5. In general we see only modest accuracy improvements when running LRC for more iterations.

Table 1 shows the wikitext-2 perplexity (PPL) and lm-eval (Gao et al., 2024) results for each method, on each model. The tasks we considered are PIQA (PQ), HellaSwag (HS), Arc-Easy (A-e), Arc-challenge (A-c), Winogrande (WG) and Lambada (LA ). We also show the average accuracy across tasks (Avg). We see that for the Phi-3 model, LRC (69.7%) recovers a substantial portion of the FP16 accuracy (72%) relative to QuaRot (64.8%). The simpler SVD approach does *not* close the accuracy gap. For the larger models Llama-2 (7 and 13 B), Llama-3 (8B) and Mixtral, we also see significant improvements.

To improve the accuracy of quantization, multiple approaches have considered use of groupsizing, where weight and activation matrices are divided into groups of size e.g. 128, and each group is scaled separately. This adds to the overall bitwidth, but improves accuracy. LRC can also be applied with groupsizing: we repeated the experiment in Table 1 in Table 2, this time applying a groups size of 128 (for activations only) to each method. Again, we see that LRC achieves multiple percentage-point improvements relative to QuaRot that are not possible with simple SVD.

**Weights only.**  To examine the effects of loss due to activation quantization, we re-ran the experiements presented above without quantizing the activations. We used the same set up as above, but we do no apply any quantizer operator in the forward pass (i.e. $Q_a$ is set to be the identity map), such that no activation quantization is performed. Table 3 shows the performances. We see that all methods are able to recover (almost) perfectly the accuracy of the original models in the W4 regime. This experiment indicates that when only the weights are quantized, there is minimal error to correct for SoTA approaches, and as a result, low-rank terms do not provide any additional improvement.

## 4.2  ABLATION STUDIES

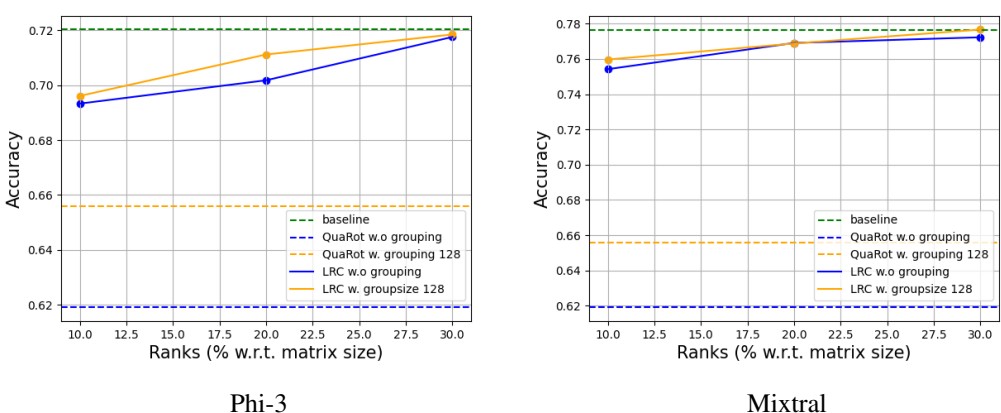

Phi-3                                        Mixtral

Figure 2: We show the effect of the rank, chosen as a percentage of the original weight matrices, on the performances of Phi-3  and Mixtral for lm-eval tasks when quantized at W4A4. We also show the effect of groupsizing activations. As baselines (dashed lines), we plot the performances of QuaRot with and without groupsizing, as well as the performance of the original models.

We perform two ablations to study the performances of LRC in different settings. We investigate the effect of the rank $k$ and the choice of the quantizer used in line 5 of Algorithm 2.

**On the effect of Rank.**  In Figure 2, we show how the choice of the rank affects our proposed algorithm LRC 1 at W4A4. We compare the average accuracy obtained across all the tasks previously described when varying the rank, chosen as a percentage of the size of weight matrices. Note that this choice of rank is adaptive to the weight matrix, and ensures that the total overhead in memory

is at most this percentage. We fix the LLM to be either Phi-3 or Mixtral and we compare the performances obtained with the QuaRot baseline where no low-rank additional matrices are added. First we observe that even with ranks equal to 10% of the original matrix size, LRC outperforms QuaRot: we also see that if we allow ranks equal to 30% of the weight-size, LRC enables close-to-lossless performance. These results hold irrespective of the use of groupsizing.

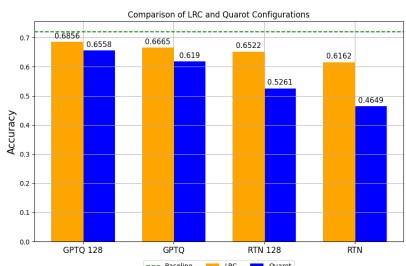

Figure 3: We show the effect of applying LRC with two quantization schemes, namely GPTQ and RTN, on the performances of Phi-3 on lm-eval tasks at W4A4.

**On the effect of Quantizer.** In Figure 3 we show how the effect of applying LRC using different quantizer in the update of $\widehat{W}$ at W4A4 on Phi-3 . In Algorithm 2, we only require to have access to a solver of the layer-wise quantization problem. By default, we use GPTQ Frantar et al. (2022) in our main experiments, but here we aim at investigating the effect of additional low-rank matrices when using other quantizers, such as simple round-to-neatest (RTN) strategies to quantize weights. We observe that, LRC is always able to improve on its baseline version where no additional low-rank matrices are added, and this gap is even more pronounced when using simpler quantization strategies like RTN.

## 5 CONCLUSION & LIMITATIONS

We have studied low-rank corrections for LLM quantization. Our main innovation is to connect the low-rank matrix to the original, pre-quantized activations, whilst processing the quantized activations with a quantized weight matrix. Our method, LRC, solves jointly for the quantization of the original weights and low rank structure to correct for errors induced by the quantization of activations. We have shown that a straight-forward approach to constructing the ranks, using SVD is not effective. Our method has the added complexity of computing activation statistics $\Sigma$, but this allows us to significantly close the accuracy gap at W4A4 by incorporating low-rank weight matrices with ranks set to 10% of the original matrix sizes. To close the gap completely, we showed that we needed ranks equal to 30% of the model size. Finally, we showed that LRC is composable with other quantization techniques, including groupsizing.

**Limitations.** In this work we have not studied the computational costs of adding low-rank computations to the forward pass. Some works have speculated that that the low-rank computation may be computable in parallel with the low-bitwidth computation: we leave such an implementation and a thorough study to future work.

We found that running our LRC procedure for multiple iterations did not comprehensively improve the performance. We found that a single iteration was often sufficient, and anecdotally we found that convergence was dependent on the damping factors used in Cholesky computations. We speculate that larger calibration set may improve the condition of the Hessians.

Finally, our work highlights that for W4A4, there is significant information lost in quantizing activations. We have followed previous works in using a scale-then-round scheme, with hyper-parameter search for the best scale. The need to perform activation quantization on-the-fly means that fast (simple!) schemes are needed. this appears to be a productive direction for future improvements.

| Method | Model | PPL | PQ | HS | A-e | A-c | WG | LA | Avg. |
|--------|-------|-----|-----|-----|-----|-----|-----|-----|------|
| FP16 | | 6.01 | 0.808 | 0.775 | 0.786 | 0.566 | 0.733 | 0.653 | 0.72 |
| QuaRot | | 7.81 | 0.77 | 0.695 | 0.74 | 0.479 | 0.635 | 0.568 | 0.648 |
| SVD | Phi-3 | 7.72 | 0.751 | 0.701 | 0.734 | 0.501 | 0.622 | 0.573 | 0.647 |
| LRC (1) | | 7.26 | **0.786** | 0.731 | 0.796 | **0.545** | **0.68** | **0.642** | **0.697** |
| LRC (5) | | **7.2** | 0.77 | **0.734** | **0.799** | 0.545 | 0.668 | 0.639 | 0.693 |
| FP16 | | 6.13 | 0.807 | 0.792 | 0.778 | 0.533 | 0.726 | 0.76 | 0.733 |
| QuaRot | | 7.78 | 0.765 | 0.74 | 0.721 | 0.441 | 0.663 | 0.704 | 0.672 |
| SVD | Llama-3 (8B) | **7.73** | 0.769 | **0.746** | 0.697 | 0.46 | 0.68 | 0.699 | 0.675 |
| LRC (1) | | 8.05 | **0.773** | 0.736 | 0.749 | 0.476 | **0.707** | 0.731 | 0.695 |
| LRC (5) | | 7.94 | 0.764 | 0.742 | **0.758** | **0.483** | 0.705 | **0.739** | **0.698** |
| FP16 | | 3.84 | 0.837 | 0.84 | 0.834 | 0.596 | 0.766 | 0.784 | 0.776 |
| QuaRot | | 4.55 | 0.813 | **0.814** | 0.794 | **0.569** | 0.726 | 0.746 | 0.744 |
| SVD | Mixtral | 4.51 | **0.817** | **0.814** | 0.802 | 0.559 | 0.726 | 0.761 | 0.746 |
| LRC (1) | | 4.42 | 0.81 | 0.801 | 0.811 | 0.561 | 0.724 | **0.818** | **0.754** |
| LRC (5) | | **4.41** | 0.801 | 0.8 | **0.813** | 0.555 | **0.736** | 0.814 | 0.753 |
| FP16 | | 5.47 | 0.791 | 0.76 | 0.745 | 0.462 | 0.691 | 0.739 | 0.698 |
| QuaRot | | 6.13 | 0.77 | 0.728 | 0.703 | 0.417 | 0.663 | 0.712 | 0.665 |
| SVD | Llama 2 (7B) | 6.12 | 0.77 | 0.729 | 0.711 | 0.436 | 0.665 | 0.717 | 0.671 |
| LRC (1) | | 5.77 | **0.776** | 0.731 | 0.726 | 0.424 | **0.676** | 0.747 | 0.68 |
| LRC (5) | | **5.75** | 0.774 | **0.733** | **0.727** | **0.439** | 0.669 | **0.748** | **0.682** |
| FP16 | | 4.88 | 0.805 | 0.794 | 0.774 | 0.491 | 0.721 | 0.767 | 0.725 |
| QuaRot | | 5.34 | 0.784 | 0.767 | 0.755 | 0.481 | 0.709 | 0.747 | 0.707 |
| SVD | Llama 2 (13B) | 5.31 | 0.792 | 0.772 | 0.755 | **0.486** | 0.699 | 0.747 | 0.709 |
| LRC (1) | | 5.09 | **0.788** | 0.77 | 0.764 | 0.482 | 0.702 | **0.781** | 0.715 |
| LRC (5) | | **5.08** | 0.786 | **0.774** | **0.769** | 0.478 | **0.706** | **0.781** | **0.716** |

Table 1: Accuracy on LLM-EVAL with weight and activation quantization (W4A4) and no group-scaling. LRC and SVD methods use low-rank matrices with 10% of the orignal matrix ranks. We have highlighted in bold the best performances among the quantized models.

| Method | Model | PPL | PQ | HS | A-e | A-c | WG | LA | Avg. |
|---|---|---|---|---|---|---|---|---|---|
| FP16 | | 6.01 | 0.808 | 0.775 | 0.786 | 0.566 | 0.733 | 0.653 | 0.72 |
| QuaRot | | 7.65 | 0.778 | 0.7 | 0.768 | 0.511 | 0.665 | 0.548 | 0.661 |
| SVD | Phi-3 | 7.54 | 0.77 | 0.696 | 0.751 | 0.52 | 0.666 | 0.555 | 0.659 |
| LRC (1) | | 7.28 | **0.786** | 0.722 | **0.815** | **0.567** | 0.693 | 0.644 | **0.704** |
| LRC (5) | | **7.25** | 0.776 | **0.728** | 0.797 | 0.539 | **0.706** | **0.65** | 0.699 |
| FP16 | | 6.13 | 0.807 | 0.792 | 0.778 | 0.533 | 0.726 | 0.76 | 0.733 |
| QuaRot | | 7.42 | 0.782 | 0.747 | 0.75 | 0.469 | 0.712 | 0.731 | 0.699 |
| SVD | Llama-3 (8B) | 7.36 | 0.779 | 0.759 | 0.762 | 0.479 | 0.72 | 0.717 | 0.703 |
| LRC (1) | | 7.03 | 0.78 | **0.762** | **0.77** | **0.505** | 0.715 | 0.764 | 0.716 |
| LRC (5) | | **7.02** | **0.783** | 0.761 | 0.766 | 0.494 | **0.735** | **0.765** | **0.717** |
| FP16 | | 3.84 | 0.837 | 0.84 | 0.834 | 0.596 | 0.766 | 0.784 | 0.776 |
| QuaRot | | 4.44 | **0.822** | 0.816 | 0.809 | **0.578** | 0.736 | 0.763 | 0.754 |
| SVD | Mixtral | 4.41 | 0.821 | **0.821** | 0.818 | 0.574 | **0.747** | 0.765 | 0.758 |
| LRC (1) | | 4.26 | 0.816 | 0.811 | 0.815 | 0.567 | 0.729 | **0.821** | 0.76 |
| LRC (5) | | **4.25** | 0.817 | 0.812 | **0.817** | 0.572 | 0.738 | 0.815 | **0.762** |
| FP16 | | 5.47 | 0.791 | 0.76 | 0.745 | 0.462 | 0.691 | 0.739 | 0.698 |
| QuaRot | | 6.12 | 0.763 | 0.725 | 0.701 | 0.41 | 0.669 | 0.715 | 0.664 |
| SVD | Llama 2 (7B) | 6.11 | 0.778 | 0.725 | 0.694 | 0.416 | 0.657 | 0.718 | 0.665 |
| LRC (1) | | 5.69 | 0.779 | **0.734** | **0.736** | **0.444** | 0.672 | 0.748 | **0.685** |
| LRC (5) | | **5.68** | **0.78** | 0.734 | 0.727 | 0.434 | **0.677** | **0.747** | 0.683 |
| FP16 | | 4.88 | 0.805 | 0.794 | 0.774 | 0.491 | 0.721 | 0.767 | 0.725 |
| QuaRot | | 5.35 | 0.782 | 0.762 | 0.758 | 0.472 | 0.702 | 0.75 | 0.705 |
| SVD | Llama 2 (13B) | 5.34 | 0.783 | 0.768 | 0.748 | 0.476 | 0.699 | 0.753 | 0.705 |
| LRC (1) | | 5.05 | 0.789 | **0.777** | **0.763** | **0.491** | **0.717** | **0.783** | **0.72** |
| LRC (5) | | **5.04** | **0.798** | 0.776 | 0.762 | **0.491** | 0.7 | 0.78 | 0.718 |

Table 2: Accuracy on LLM-EVAL with weight and activation quantization (W4A4). For each method we use a groupsize of 128 for activations. We have highlighted in bold the best performances among the quantized models.

| Method | Model | Size | PPL | PQ | HS | A-e | A-c | WG | LA | Avg. |
|---|---|---|---|---|---|---|---|---|---|---|
| FP16 | | 6.75 | 6.01 | 0.808 | 0.775 | 0.786 | 0.566 | 0.733 | 0.653 | 0.72 |
| QuaRot | | 1.69 | 6.3 | 0.804 | 0.756 | **0.781** | 0.561 | 0.719 | 0.642 | 0.711 |
| SVD | Phi-3 | 2.59 | 6.24 | **0.808** | 0.759 | 0.779 | **0.567** | 0.727 | 0.646 | **0.714** |
| LRC | | 2.59 | **6.21** | 0.805 | **0.76** | 0.772 | 0.558 | 0.723 | 0.641 | 0.71 |
| FP16 | | 13 | 6.13 | 0.807 | 0.792 | 0.778 | 0.533 | 0.726 | 0.76 | 0.733 |
| QuaRot | | 3.25 | 6.55 | **0.805** | 0.779 | **0.774** | **0.519** | **0.742** | 0.74 | **0.727** |
| SVD | Llama-3 (8B) | 4.95 | 6.49 | 0.799 | **0.783** | 0.765 | 0.508 | 0.738 | **0.749** | 0.724 |
| LRC | | 4.95 | **6.47** | 0.8 | 0.78 | 0.761 | 0.51 | 0.731 | 0.747 | 0.722 |
| FP16 | | 86.5 | 3.84 | 0.837 | 0.84 | 0.834 | 0.596 | 0.766 | 0.784 | 0.776 |
| QuaRot | | 21.6 | 3.98 | 0.836 | 0.836 | 0.825 | **0.593** | 0.757 | 0.781 | 0.771 |
| SVD | Mixtral | 32.1 | 3.96 | 0.835 | 0.837 | 0.832 | **0.593** | **0.766** | **0.787** | **0.775** |
| LRC | | 32.1 | **3.95** | **0.84** | **0.839** | **0.825** | **0.593** | 0.754 | 0.783 | 0.772 |

Table 3: Accuracy on LLM-EVAL with weight only quantization. We have highlighted in bold the best performances among the quantized models. The size is given in GB.

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

# APPENDIX

## A  ADDITIONAL BACKGROUND

**GPTQ Algorithm.**   The GPTQ algorithm, introduced by Frantar et al. (2022), is a post-training quantization technique designed to efficiently reduce the precision of weights in large language models (LLMs) while maintaining their performance. To achieve this, the authors propose to approximate a solution of the layer-wise quadratic approximation problem defined as:

$$\min_{\widehat{\boldsymbol{W}} \in \mathcal{C}(b) \cap \mathbb{R}^{d^{\text{out}} \times d^{\text{in}}}} \mathcal{L}_{\text{q}}(\widehat{\boldsymbol{W}}) := \|\boldsymbol{W}\boldsymbol{X} - \widehat{\boldsymbol{W}}\boldsymbol{X}\|_2^2 \,,$$

where $\boldsymbol{W}$ is the original weight matrix, and $\mathcal{C}(b)$ is the constraint set of matrices admitting a certain bit per weight $b > 0$ precision. The main difficulty of solving exactly this optimization problem resides in the constraint set $\mathcal{C}(b)$, making the problem non-convex. To approximate a solution, (Frantar et al., 2022) propose to improve the computational scheme of the greedy approach originally proposed by LeCun et al. (1989) for pruning, and then adapted for quantization in (Frantar & Alistarh, 2022), by removing the ordering in the greedy quantization process, and applying the algorithm in parallel over multiple columns.

**Cholesky Factorization.**   Cholesky factorization is a numerical method used to decompose a symmetric positive-definite matrix (PD) into the product of a lower triangular matrix with positive diagonal coefficients and its transpose. This technique is particularly useful in solving systems of linear equations, performing matrix inversion, and computing the determinant of a matrix. More formally given $\boldsymbol{\Sigma}$ a symmetric PD matrix, there exists a unique lower triangular matrix $\boldsymbol{L}$ such that

$$\boldsymbol{\Sigma} = \boldsymbol{L}\boldsymbol{L}^\top \,.$$

To compute the Cholesky factor $\boldsymbol{L}$, one can rely on the Cholesky Algorithm which is a modified version of the Gaussian elimination and requires $\mathcal{O}(n^3)$ FLOPs where $n$ is the size of $\boldsymbol{\Sigma}$.

## B  ALGORITHMS

**On the Update of $\widehat{\boldsymbol{W}}$.**   According to Proposition 3.1, updating $\widehat{\boldsymbol{W}}$ according to equation 3, is equivalent to solving equation 5, which can be approximated by using any solvers designed for equation 1 such as (Frantar et al., 2022; Lin et al., 2024; Egiazarian et al., 2024). In practice, we use the GPTQ algorithm (Frantar et al., 2022) that only requires access to the target weight matrix $\widetilde{\boldsymbol{W}}^{(t)}$ and the covariance matrix $\boldsymbol{Y}\boldsymbol{Y}^\top$.

---

**Algorithm 2** Update-Quant$(\boldsymbol{W}, \boldsymbol{U}, \boldsymbol{V}, \boldsymbol{Y}\boldsymbol{Y}^\top, \boldsymbol{X}\boldsymbol{Y}^\top, b)$

---

1: **Input:** Original weight matrix $\boldsymbol{W}$, low-rank weight matrices $\boldsymbol{U}, \boldsymbol{V}$, covariance matrix $\boldsymbol{Y}\boldsymbol{Y}^\top$, cross-covariance matrix $\boldsymbol{X}\boldsymbol{Y}^\top$, and the bit precision $b$.
2: **Output:** Quantized weight matrix $\widehat{\boldsymbol{W}}$.
3: $\boldsymbol{L}_{\boldsymbol{Y}} \leftarrow \text{Cholesky}(\boldsymbol{Y}\boldsymbol{Y}^\top)$
4: $\widetilde{\boldsymbol{W}} \leftarrow (\boldsymbol{W} - \boldsymbol{U}\boldsymbol{V}^\top)\boldsymbol{X}\boldsymbol{Y}^\top (\boldsymbol{L}_{\boldsymbol{Y}}^\top)^{-1}\boldsymbol{L}_{\boldsymbol{Y}}^{-1}$
5: $\widehat{\boldsymbol{W}} \leftarrow \text{GPTQ}(\widetilde{\boldsymbol{W}}, \boldsymbol{Y}\boldsymbol{Y}^\top, b)$ using Alg. of Frantar et al. (2022)
6: **return** $\widehat{\boldsymbol{W}}$

---

*Remark* B.1.  In line 3 of Algorithm 2, we use the Cholesky decomposition of the covariance $\boldsymbol{Y}\boldsymbol{Y}^\top$ to compute $(\boldsymbol{W} - \boldsymbol{U}\boldsymbol{V}^\top)\boldsymbol{X}\boldsymbol{Y}^\top(\boldsymbol{Y}\boldsymbol{Y}^\top)^{-1}$ for better numerical stability.

**On the Update of $\boldsymbol{U}, \boldsymbol{V}$.**   While solving exactly equations 1, or 5 is still an open question due to the non-convexity of the constraints, obtaining the update for $\boldsymbol{U}, \boldsymbol{V}$, that is solving equation 4, can be done in closed form, as shown in the Proposition 3.3.

Note that in line 8 of the Algorithm 3, we denote $\text{eig}_k(\cdot)$ the operator that returns the $k$ first unit eigenvectors of a symmetric matrix ranked in the decreasing order w.r.t their eigenvalues.

---

**Algorithm 3** Update-LR($\boldsymbol{W}, \widehat{\boldsymbol{W}}, \boldsymbol{X}\boldsymbol{X}^\top, \boldsymbol{X}\boldsymbol{Y}^\top, k$)

---

1: **Input:** Original weight matrix $\boldsymbol{W}$, quantized weight matrix $\widehat{\boldsymbol{W}}$, covariance matrix $\boldsymbol{X}\boldsymbol{X}^\top$, cross-covariance matrix $\boldsymbol{X}\boldsymbol{Y}^\top$, and the rank $k$.
2: **Output:** Low-rank weight matrices $\boldsymbol{U}, \boldsymbol{V}$.
3: $\boldsymbol{\Sigma_1} \leftarrow \boldsymbol{W}\boldsymbol{X}\boldsymbol{X}^\top\boldsymbol{W}^\top$
4: $\boldsymbol{\Sigma_3} \leftarrow \widehat{\boldsymbol{W}}\boldsymbol{Y}\boldsymbol{X}^\top\boldsymbol{W}^\top + \boldsymbol{W}\boldsymbol{X}\boldsymbol{Y}^\top\widehat{\boldsymbol{W}}^\top$
5: $\boldsymbol{L_X} \leftarrow \text{Cholesky}(\boldsymbol{X}\boldsymbol{X}^\top), \quad \boldsymbol{S} \leftarrow \boldsymbol{L_X}^{-1}\boldsymbol{X}\boldsymbol{Y}^\top\widehat{\boldsymbol{W}}^\top$
6: $\boldsymbol{\Sigma_2} \leftarrow \boldsymbol{S}^\top\boldsymbol{S}$
7: $\boldsymbol{\Sigma} \leftarrow \boldsymbol{\Sigma_1} + \boldsymbol{\Sigma_2} - \boldsymbol{\Sigma_3}$
8: $\boldsymbol{U} \leftarrow \text{eig}_k(\boldsymbol{\Sigma}), \quad \boldsymbol{V} \leftarrow \left[\boldsymbol{W}^\top - (\boldsymbol{X}\boldsymbol{X}^\top)^{-1}\boldsymbol{X}\boldsymbol{Y}^\top\widehat{\boldsymbol{W}}^\top\right]\boldsymbol{U}$
9: **return** $\boldsymbol{U}, \boldsymbol{V}$

---

**Initialization.** To initialize our algorithm, that is to instantiate $\boldsymbol{U}^{(0)}$ and $\boldsymbol{V}^{(0)}$, we propose to consider the optimization problem defined in equation 7, which can be solved in closed form as shown in the Proposition 3.4.

---

**Algorithm 4** Init-LR($\boldsymbol{W}, \boldsymbol{X}\boldsymbol{X}^\top, \boldsymbol{Y}\boldsymbol{Y}^\top, \boldsymbol{X}\boldsymbol{Y}^\top, k$)

---

1: **Input:** Original weight matrix $\boldsymbol{W}$, covariance matrices $\boldsymbol{X}\boldsymbol{X}^\top, \boldsymbol{Y}\boldsymbol{Y}^\top$, cross-covariance matrix $\boldsymbol{X}\boldsymbol{Y}^\top$, and the rank $k$.
2: **Output:** Low-rank weight matrices $\boldsymbol{U}, \boldsymbol{V}$.
3: $\boldsymbol{\Sigma_1} \leftarrow \boldsymbol{W}\boldsymbol{X}\boldsymbol{X}^\top\boldsymbol{W}^\top$
4: $\boldsymbol{L_Y} \leftarrow \text{Cholesky}(\boldsymbol{Y}\boldsymbol{Y}^\top), \quad \boldsymbol{S} \leftarrow \boldsymbol{L_Y}^{-1}\boldsymbol{Y}\boldsymbol{X}^\top\boldsymbol{W}^\top$
5: $\boldsymbol{\Sigma_2} \leftarrow \boldsymbol{S}^\top\boldsymbol{S}$
6: $\boldsymbol{\Sigma}_{\text{init}} \leftarrow \boldsymbol{\Sigma_1} - \boldsymbol{\Sigma_2}$
7: $\boldsymbol{U} \leftarrow \text{eig}_k(\boldsymbol{\Sigma}_{\text{init}}), \quad \boldsymbol{V} \leftarrow \boldsymbol{W}^\top\boldsymbol{U}$
8: **return** $\boldsymbol{U}, \boldsymbol{V}$

---

**LRC Algorithm.** We are now ready to present the full LRC algorithm, as presented in the main text in Alg. 1.

---

**Algorithm 5** LRC($\boldsymbol{W}, \boldsymbol{X}, b, a, k$)

---

1: **Input:** Original weight matrix $\boldsymbol{W}$, activation $\boldsymbol{X}$, the bit precision for weights $b$, the bit precision for activation $a$, the rank $k$, and number of iterations $T$.
2: **Output:** Quantized weight matrix $\widehat{\boldsymbol{W}}$, low-rank weight matrices $\boldsymbol{U}, \boldsymbol{V}$
3: $\boldsymbol{\Sigma}_x \leftarrow \boldsymbol{X}\boldsymbol{X}^\top + \epsilon_x\text{I}_{d^{\text{in}}}$
4: $\boldsymbol{Y} \leftarrow Q_a(\boldsymbol{X}), \quad \boldsymbol{\Sigma}_y \leftarrow \boldsymbol{Y}\boldsymbol{Y}^\top + \epsilon_y\text{I}_{d^{\text{in}}}$
5: $\boldsymbol{\Sigma}_{xy} \leftarrow \boldsymbol{X}\boldsymbol{Y}^\top$
6: $\boldsymbol{U}, \boldsymbol{V} \leftarrow \text{Init-LR}(\boldsymbol{W}, \boldsymbol{\Sigma}_x, \boldsymbol{\Sigma}_y, \boldsymbol{\Sigma}_{xy}, k), \quad$ using Alg. 4
7: **for** $t = 1, ..., T$ **do**
8: $\quad \widehat{\boldsymbol{W}} \leftarrow \text{Update-Quant}(\boldsymbol{W}, \boldsymbol{U}, \boldsymbol{V}, \boldsymbol{\Sigma}_y, \boldsymbol{\Sigma}_{xy}, b), \quad$ using Alg. 2
9: $\quad \boldsymbol{U}, \boldsymbol{V} \leftarrow \text{Update-LR}(\boldsymbol{W}, \widehat{\boldsymbol{W}}, \boldsymbol{\Sigma}_x, \boldsymbol{\Sigma}_{xy}, k), \quad$ using Alg. 3
10: **end for**
11: **return** $\widehat{\boldsymbol{W}}, \boldsymbol{U}, \boldsymbol{V}$

---

## C  ADDITIONAL EXPERIMENTS

### C.1  EFFECT OF THE CALIBRATION DATASET

In this section, we investigate the effect of the calibration dataset selection on the performance of LRC. Our observations indicate that the choice of the calibration dataset does not significantly affect

the performance of the quantized models on downstream tasks. In tables 4, 5, we compare the LRC performance with a rank set to 10% of the original size on Phi-3 at W4A4.

| Dataset | Avg. | A-c | A-e | HS | LA | PQ | WG |
|---------|------|------|------|------|------|------|------|
| Alpaca | 0.7024 | 0.5478 | 0.7795 | 0.7234 | 0.6553 | 0.7884 | 0.7198 |
| wikitext2 | 0.7 | 0.5452 | 0.779 | 0.7264 | 0.6505 | 0.784 | 0.7151 |

Table 4: Accuracy of LRC on downstram tasks when using either the Wikitext2 or Alpaca dataset with groupsizing (128) on activations.

| Dataset | Avg. | A-c | A-e | HS | LA | PQ | WG |
|---------|------|------|------|------|------|------|------|
| Alpaca | 0.6891 | 0.5273 | 0.7626 | 0.699 | 0.6588 | 0.7737 | 0.7135 |
| Wikitext2 | 0.6917 | 0.5341 | 0.7782 | 0.713 | 0.6511 | 0.7835 | 0.6906 |

Table 5: Accuracy of LRC on downstram tasks when using either the Wikitext2 or Alpaca dataset without groupsizing.

## C.2 LATENCY OF LRC

In our experiments presented in Tables 2, 1, 3, we settled on setting the rank to 10% of the original size which incurs an additional memory 13% of the original model (see the sizes reported in Table 3). Therefore, we are effectively at 6.08 bits $(4 + 0.13 * 16)$.

In this section, we set up a simple timing experiment on an Nvidia A100 device to time the cost of a forward pass. We use a batch size of 32, sequence length of 2048, and matrix sizes from the Llama model series. We used Cutlass to implement a basic int4 kernel. We timed the cost of quantizing the activations, computing the int4 kernel, computing the low-rank matmul in fp16, and adding the results. Our pytorch module looks like this:

```python
baseline_mod = torch.nn.Linear(feature_dim_in, feature_dim_out, bias=
    False).cuda().to(torch.float16)
class Int4Lowrank(torch.nn.Module):
    def __init__(self):
        super().__init__()
        self.quant = Quantizer(input_clip_ratio=1.0)
        self.U = torch.nn.Linear(feature_dim_in, ranks, bias=False).to(
            torch.float16)
        self.V = torch.nn.Linear(ranks, feature_dim_out, bias=False).to(
            torch.float16)
        self.lin_4bit = Linear4bit.from_float(baseline_mod, weight_scales
            =s_w)
    @torch.compile()
    def forward(self, x):
        return self.lin_4bit(self.quant(x)) + self.V(self.U(x))
```

Listing 1: Python code snippet of a naive implementation of LRC.

Here are the timings of this simple layer, with warmup, repeated 100x. Matrix sizes taken from the Llama family.

We see that adding low-rank weight matrices does increase the latency of these operations as expected, though we still retain speedup relative to full FP16. In each row of each table, we have highlighted the choice of ranks that is above (next power 2) the 10% factor we used in the main experiments in the paper.

We have included numbers from very small ranks to emphasize a limitation of this experiment: even with a very small number of ranks added (128) there is latency loss. This implies that data movement is important, and that a fused kernel could improve latency.

| ranks | matrix dim | time (ms) | speedup over fp16 |
|-------|------------|-----------|-------------------|
| 0 | 11008x4096 | 13.89 +- 0.23 | 1.97 |
| 128 | 11008x4096 | 18.04 +- 0.16 | 1.52 |
| 256 | 11008x4096 | 19.019 +- 0.21 | 1.45 |
| **512** | 11008x4096 | 21.284 +- 0.2 | **1.29** |
| 1024 | 11008x4096 | 25.87 +- 0.26 | 1.06 |

Table 6: Performance metrics for matrix dimension 11008x4096

| ranks | matrix dim | time (ms) | speedup over fp16 |
|-------|------------|-----------|-------------------|
| 0 | 13824x5120 | 20.15 +- 0.03 | 2.03 |
| 128 | 13824x5120 | 25.15 +- 0.09 | 1.63 |
| 256 | 13824x5120 | 26.25 +- 0.05 | 1.56 |
| **512** | 13824x5120 | 29.140 +- 0.08 | **1.40** |
| 1024 | 13824x5120 | 34.77 +- 0.15 | 1.18 |

Table 7: Performance metrics for matrix dimension 13824x5120

This experiment is also limited in that it does not account from groupsizing, which would make the addition of low-rank matrices more appealing in terms of latency since int4 operations would themselves be reduced in speed.

### C.3 CLOSING THE ACCURACY GAP WITH LRC AT W4A4

In addition to the experiment presented in Figure 2, we also investigate the effect of the rank in LRC on Llama-3 (8B). As previously observed for Phi-3 and Mixtral, we observe that when the rank is set to 30% of the original size, LRC is able to recover the performances of the original models. In Figure 4, we show how to choice of the rank impacts the performances of the LLM on the downstream tasks considered in this work. Additionally, we report in tables 9, 10 detailed scores obtained by quantized LLMs with LRC using 30% additional ranks.

## D  PROOFS

### D.1  PROOF OF PROPOSITION 3.1

Let us first denote the objective function defined in equation 3:

$$\mathcal{L}_q(\widehat{W}) := \|WX - \widehat{W}Y - UV^\top X\|_2^2$$

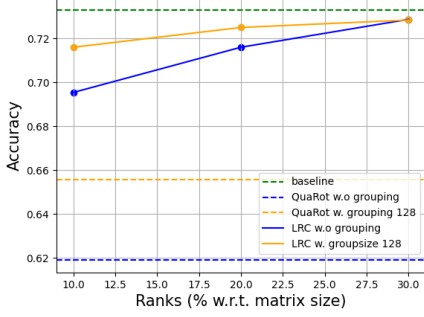

Figure 4: We show the effect of the rank, chosen as a percentage of the original weight matrices, on the performances of Llama-3 (8B) for lm-eval tasks when quantized at W4A4. We also show the effect of groupsizing activations. As baselines (dashed lines), we plot the performances of QuaRot with and without groupsizing, as well as the performance of the original model.

| ranks | matrix dim | time (ms) | speedup over fp16 |
|---|---|---|---|
| 0 | 28672x8192 | 54.83 +- 0.71 | 2.44 |
| 128 | 28672x8192 | 64.40 +- 0.17 | 2.07 |
| 256 | 28672x8192 | 66.77 +- 0.18 | 2.0 |
| 512 | 28672x8192 | 72.03 +- 0.2 | 1.86 |
| **1024** | 28672x8192 | 82.98 +- 0.40 | **1.62** |

Table 8: Performance metrics for matrix dimension 28672x8192

| Method | Model | Size | PPL | PQ | HS | A-e | A-c | WG | LA | Avg. |
|---|---|---|---|---|---|---|---|---|---|---|
| FP16 | Phi-3 | 6.75 | 6.01 | 0.808 | 0.775 | 0.786 | 0.566 | 0.733 | 0.653 | 0.72 |
| LRC | | 4.39 | 6.4 | 0.801 | 0.746 | 0.808 | 0.579 | 0.721 | 0.649 | 0.718 |
| FP16 | Llama-3 (8B) | 13 | 6.13 | 0.807 | 0.792 | 0.778 | 0.533 | 0.726 | 0.76 | 0.733 |
| LRC | | 8.35 | 6.72 | 0.799 | 0.771 | 0.784 | 0.503 | 0.744 | 0.771 | 0.729 |
| FP16 | Mixtral | 86.5 | 3.84 | 0.837 | 0.84 | 0.834 | 0.596 | 0.766 | 0.784 | 0.776 |
| LRC | | 53 | 4.12 | 0.821 | 0.825 | 0.827 | 0.584 | 0.759 | 0.818 | 0.772 |

Table 9: Accuracy on LLM-EVAL with weight and activation quantization (W4A4) and no group-scaling. LRC method uses low-rank matrices with 30% of the orignal matrix ranks. The size is given in GB.

where we omit the superscript $t$ of $U, V$ for simplicity. By simply developing the objective and discarding the constant term (i.e. those independent of $\widehat{W}$), we can reformulate the objective function as:

$$\langle \widehat{W}, \widehat{W}\Sigma_y \rangle - 2[\langle \widehat{W}, (W - UV^\top)XY^\top \rangle]$$
$$= \langle \widehat{W}, \widehat{W}\Sigma_y \rangle - 2[\langle \widehat{W}, (W - UV^\top)XY^\top\Sigma_y\Sigma_y^{-1} \rangle]$$

where $\Sigma_y := YY^\top$, and the second equality holds under the the assumption that $Y$ is full rank with $n \geq d_{\text{in}}$. Then by denoting

$$\widetilde{W} := (W - UV^\top)XY^\top\Sigma_y^{-1}$$

we obtain that the objective, as function of $\widehat{W}$ is equivalent to

$$\langle \widehat{W} - \widetilde{W}, (\widehat{W} - \widetilde{W})\Sigma_y \rangle = \|\widehat{W}Y - \widetilde{W}Y\|_2^2$$

which conclude the proof.

### D.2 PROOF OF PROPOSITION 3.3

Let us first denote the objective function defined in equation 4 as

$$\mathcal{L}_{\text{lr}}(U, V) := \|WX - \widehat{W}Y - UV^\top X\|_2^2$$

| Method | Model | Size | PPL | PQ | HS | A-e | A-c | WG | LA | Avg. |
|---|---|---|---|---|---|---|---|---|---|---|
| FP16 | Phi-3 | 6.75 | 6.01 | 0.808 | 0.775 | 0.786 | 0.566 | 0.733 | 0.653 | 0.72 |
| LRC | | 4.39 | 6.31 | 0.796 | 0.752 | 0.8 | 0.574 | 0.73 | 0.658 | 0.719 |
| FP16 | Llama-3 (8B) | 13 | 6.13 | 0.807 | 0.792 | 0.778 | 0.533 | 0.726 | 0.76 | 0.733 |
| LRC | | 8.35 | 6.58 | 0.794 | 0.775 | 0.788 | 0.517 | 0.725 | 0.772 | 0.728 |
| FP16 | Mixtral | 86.5 | 3.84 | 0.837 | 0.84 | 0.834 | 0.596 | 0.766 | 0.784 | 0.776 |
| LRC | | 53 | 4.04 | 0.831 | 0.826 | 0.835 | 0.596 | 0.762 | 0.811 | 0.777 |

Table 10: Accuracy on LLM-EVAL with weight and activation quantization (W4A4). LRC method use low-rank matrices with 30% of the orignal matrix ranks and a groupsize of 128 for activations. The size is given in GB.

where we omit the superscript $t$ of $\widehat{W}$ for simplicity. Let us now write the first order condition for $V$. Indeed we obtain that:

$$\frac{\partial \mathcal{L}_{\mathrm{lr}}}{\partial V} = 0 \quad \text{which gives} \quad U^\top U V^\top \Sigma_x = U^\top [W \Sigma_x - \widehat{W} Y X^\top]$$

where $\Sigma_x := X X^\top$ Under the assumption that $X$ is full rank with $n \geq d_{\mathrm{in}}$, we deduce that

$$U^\top U V^\top = U^\top [W - \widehat{W} Y X^\top \Sigma_x^{-1}]$$

which always admits a solution. Then by denoting $(U^\top U)^{-1}$ the Moore–Penrose inverse of $U^\top U$, we obtain that:

$$V^\top = (U^\top U)^{-1} U^\top [W - \widehat{W} Y X^\top \Sigma_x^{-1}]$$

Then by plugging this expression into the original objective, we obtain the following equivalent optimization problem:

$$\min_{U \in \mathbb{R}^{d^{\mathrm{out}} \times k}} \|W X - \widehat{W} Y - U(U^\top U)^{-1} U^\top [W - \widehat{W} Y X^\top \Sigma_x^{-1}] X\|_F^2$$

and as $\{U(U^\top U)^{-1} U^\top : \ U \in \mathbb{R}^{d^{\mathrm{out}} \times k}\}$ spans the space of orthogonal projection onto subspaces of dimension at most $k$, we can reparameterize the optimization problem as:

$$\min_{U \in \mathbb{R}^{d^{\mathrm{out}} \times k} \cap \mathcal{O}} \|W X - \widehat{W} Y - U U^\top [W - \widehat{W} Y X^\top \Sigma_x^{-1}] X\|_2^2$$

Then by developing the objective and discarding the constant terms (i.e. those independent of $U$), we obtain the following equivalent objective:

$$\mathrm{Tr}(U^\top [\widehat{W} Y X^\top W^\top + W X Y^\top \widehat{W}^\top] U) - \mathrm{Tr}(U^\top W \Sigma_x W^\top U) - \mathrm{Tr}(U^\top \widehat{W} Y X^\top \Sigma_x^{-1} X Y^\top \widehat{W}^\top U)$$

Therefore minimizing the above objective w.r.t $U \in \mathbb{R}^{d^{\mathrm{out}} \times k} \cap \mathcal{O}$, is equivalent to maximize w.r.t $U \in \mathbb{R}^{d^{\mathrm{out}} \times k} \cap \mathcal{O}$ the following objective:

$$\mathrm{Tr}(U^\top (\Sigma_1 + \Sigma_2 - \Sigma_3) U)$$

where

$$\Sigma_1 := W X X^\top W^\top, \ \ \Sigma_2 := \widehat{W} Y X^\top (X X^\top)^{-1} X Y^\top \widehat{W}^\top, \ \text{and}$$
$$\Sigma_3 := \widehat{W} Y X^\top W^\top + W X Y^\top \widehat{W}^\top .$$

which is exactly the optimization problem defined in Proposition 3.3. Finally observe $\Sigma := \Sigma_1 + \Sigma_2 - \Sigma_3$ is symmetric but not necessarily positive semi-definite. However, observe that for any symmetric matrix $\Sigma \in \mathbb{R}^{d \times d}$, $U \in \mathbb{R}^{d \times k} \cap \mathcal{O}$ and $\alpha > 0$, we have:

$$\mathrm{Tr}(U^\top \Sigma U) = \mathrm{Tr}(U^\top (\Sigma + \alpha \mathrm{I}_d) U) - \alpha \mathrm{Tr}(U^\top U) = \mathrm{Tr}(U^\top (\Sigma + \alpha \mathrm{I}_d) U) + k\alpha$$

So by taking $\alpha$ sufficiently large such that $\Sigma + \alpha \mathrm{I}_d$ is P.D., we deduces that $U$ can be chosen to be the $k$ first unit eigenvectors of $\Sigma + \alpha \mathrm{I}_d$, which are the same as $\Sigma$, and that concludes the proof.

### D.3 PROOF OF PROPOSITION 3.4

Observe that we can rewrite the optimization problem as:

$$\min_{U \in \mathbb{R}^{d^{\mathrm{out}} \times k}, V \in \mathbb{R}^{d^{\mathrm{in}} \times k}} \min_{\widetilde{W} \in \mathbb{R}^{d^{\mathrm{out}} \times d^{\mathrm{in}}}} \mathcal{L}_{\mathrm{qlr}}(\widetilde{W}, U, V) .$$

Now given $U \in \mathbb{R}^{d^{\mathrm{out}} \times k}, V \in \mathbb{R}^{d^{\mathrm{in}} \times k}$, we can derive the first order condition of the inner optimization problem, that is:

$$\frac{\partial \mathcal{L}_{\mathrm{qlr}}}{\partial \widetilde{W}} = 0 \quad \text{which gives} \quad \widetilde{W} \Sigma_y = [W - U V^\top] X Y^\top$$

where $\Sigma_y := Y Y^\top$. Now under the assumption that $Y \in \mathbb{R}^{d_{\mathrm{in}} \times n}$ is full rank where $n \geq d_{\mathrm{in}}$, we obtain that

$$\widetilde{\boldsymbol{W}} = [\boldsymbol{W} - \boldsymbol{U}\boldsymbol{V}^T]\boldsymbol{X}\boldsymbol{Y}^T\boldsymbol{\Sigma}_y^{-1}$$

from which we deduce the equivalent optimization problem:

$$\min_{\boldsymbol{U}\in\mathbb{R}^{d^{\text{out}}\times k}, \boldsymbol{V}\in\mathbb{R}^{d^{\text{in}}\times k}} \|(\boldsymbol{W} - \boldsymbol{U}\boldsymbol{V}^\top)\tilde{\boldsymbol{X}}\|_2^2$$

where $\tilde{\boldsymbol{X}} := \boldsymbol{X} - \boldsymbol{X}\boldsymbol{Y}^\top\boldsymbol{\Sigma}_y^{-1}\boldsymbol{Y}$. Again, by fixing $\boldsymbol{U}$ and by deriving the first order condition for $\boldsymbol{V}$, we obtain that:

$$\boldsymbol{U}^\top\boldsymbol{U}\boldsymbol{V}^\top\tilde{\boldsymbol{X}}\tilde{\boldsymbol{X}}^\top = \boldsymbol{U}^\top\boldsymbol{W}\tilde{\boldsymbol{X}}\tilde{\boldsymbol{X}}^\top$$

Assuming that $\tilde{\boldsymbol{X}}$ is full rank, we recover the normal equation

$$\boldsymbol{U}^\top\boldsymbol{U}\boldsymbol{V}^\top = \boldsymbol{U}^\top\boldsymbol{W}$$

which always admit a solution. Then by denoting $(\boldsymbol{U}^\top\boldsymbol{U})^{-1}$ the Moore–Penrose inverse of $\boldsymbol{U}^\top\boldsymbol{U}$, we obtain that:

$$\boldsymbol{V}^\top = (\boldsymbol{U}^\top\boldsymbol{U})^{-1}\boldsymbol{U}^\top\boldsymbol{W}$$

Plugging back this expression to the previous objective leads to the following optimization problem:

$$\min_{\boldsymbol{U}\in\mathbb{R}^{d^{\text{out}}\times k}} \|(\boldsymbol{I}_d - \boldsymbol{U}(\boldsymbol{U}^\top\boldsymbol{U})^{-1}\boldsymbol{U}^\top)\boldsymbol{W}\tilde{\boldsymbol{X}}\|_2^2$$

and by denoting $\boldsymbol{O} := \boldsymbol{W}\tilde{\boldsymbol{X}}$, we recover the PCA of $\boldsymbol{O}$, that is:

$$\min_{\boldsymbol{U}\in\mathbb{R}^{d^{\text{out}}\times k}} \|(\boldsymbol{I}_d - \boldsymbol{U}(\boldsymbol{U}^\top\boldsymbol{U})^{-1}\boldsymbol{U}^\top)\boldsymbol{O}\|_2^2$$

as $\{\boldsymbol{U}(\boldsymbol{U}^\top\boldsymbol{U})^{-1}\boldsymbol{U}^\top : \boldsymbol{U}\in\mathbb{R}^{d^{\text{out}}\times k}\}$ spans the space of orthogonal projection onto subspaces of dimension at most $k$. Now observe that

$$\boldsymbol{O}\boldsymbol{O}^\top = \boldsymbol{W}\boldsymbol{X}[I_n - \boldsymbol{Y}^\top(\boldsymbol{Y}\boldsymbol{Y}^\top)^{-1}\boldsymbol{Y}]\boldsymbol{X}^\top\boldsymbol{W}^\top = \boldsymbol{\Sigma}_{\text{init}}$$

which conclude the proof.

