# OpenReview forum: "Low-Rank Correction for Quantized LLMs"
_ICLR.cc/2025/Conference — Submitted to ICLR 2025_

### Official Review · Reviewer_LQqC · 2024-11-02

**Soundness:** 2
**Presentation:** 3
**Contribution:** 2
**Rating:** 3
**Confidence:** 4

**Summary:**

This work presents a low-rank approach aimed at enhancing the performance of post-training quantization. Specifically, it refines the Low-Rank Correction method by incorporating selected low-rank matrices in full precision during the forward pass to help reduce quantization errors. The experimental results indicate competitive performance, showing improvements over current methods, espeically at W4A4 quantization levels.

**Strengths:**

1. The studied problem is important.
2. This paper provides a new method of low rank correction to consider both weights and activations are quantized situation.
3. This paper shows a good experimatal performance on low bit quantazation W4A4.

**Weaknesses:**

1. The paper does not provide sufficient proof of the correctness of its algorithm. There is no error analysis for the approximation methods presented. While the authors detail their approach for solving Equations 3 and 4, they do not demonstrate that these solutions are equivalent to Equation 2, which is the main problem of interest.
2. The theoretical analysis provided by the authors is overly simplified. The assumptions of full rank in Propositions 3.3 and 3.4 are questionable. Empirical studies have suggested that activations in large transformer models often exhibit a structure that can be approximated as low rank, making these assumptions potentially unsuitable.
3. The paper does not include an analysis of the time complexity associated with adding low-rank weights. This omission leaves a gap in understanding the computational implications of the proposed method.
4. The contribution and novelty of the paper are somewhat limited. While the authors improve the accuracy of post-training quantization through low-rank correction, additional analyses on aspects such as time complexity, convergence, and error bounds would enhance the paper’s competitiveness and impact.

**Questions:**

See weakness.

---

> ### Author Response · Authors · 2024-11-22
> **Rebuttal by Authors**
>
> >The paper does not provide sufficient proof of the correctness of its algorithm. There is no error analysis for the approximation methods presented. While the authors detail their approach for solving Equations 3 and 4, they do not demonstrate that these solutions are equivalent to Equation 2, which is the main problem of interest.
>
> We thank the reviewer for poiting out this omission. Please find in Appendix C all the proofs of the results presented in our manuscript.
>
> >The theoretical analysis provided by the authors is overly simplified. The assumptions of full rank in Propositions 3.3 and 3.4 are questionable. Empirical studies have suggested that activations in large transformer models often exhibit a structure that can be approximated as low rank, making these assumptions potentially unsuitable.
>
> Thank you for your comment. We would like to clarify that although we have presented the 'invertible' case for simplicity, our approach is readily extendable to scenarios where a regularization term is applied to the covariance matrices. This extension is precisely the case we consider in practice, as detailed in the paragraph 'Numerical Stability' (line 303).
>
> >The paper does not include an analysis of the time complexity associated with adding low-rank weights. This omission leaves a gap in understanding the computational implications of the proposed method.
>
> We thank the reviewer for this important remark. Currently we are only emulating the implementation of our approach and do not leverage a specific cuda kernel adapted to our computational scheme in order to perform the low-rank correction. However, for the sake of evaluation, we have added an experiment evaluating the speedup obtained with a naive cuda kernel implementing our approach and compare it against the full precision version.  This experiment is presented at the begining of the rebuttal.
>
> >The contribution and novelty of the paper are somewhat limited. While the authors improve the accuracy of post-training quantization through low-rank correction, additional analyses on aspects such as time complexity, convergence, and error bounds would enhance the paper’s competitiveness and impact.
>
> We concur that these additional theoretical considerations may indeed bolster the approach. Although our proposed algorithm is underpinned by a robust theoretical framework, we contend that the primary value of our work lies in its practical relevance to the community, rather than its theoretical contributions.

---

> > ### Comment · Reviewer_LQqC · 2024-11-25
> >
> > Thank you for your detailed reply. After reviewing the response, I have the following concerns:
> >
> > 1. **Lack of theoretical rigor**: The main text does not include a formal proof of the algorithm's correctness. At a minimum, the paper should include a theoretical justification and summarize it in a clear theorem.
> >
> > 2. **Limited novelty**: The contributions lack sufficient innovation to meet the standards of ICLR. The approach appears incremental and does not offer significant advancements over existing methods.
> >
> > Based on these issues, I find the paper does not meet the standards for acceptance at ICLR. I encourage the authors to address these concerns and strengthen both the theoretical and empirical aspects in future submissions.

---

> > > ### Author Response · Authors · 2024-11-25
> > > **Thanks for your prompt reaction**
> > >
> > > Thanks for your reaction. Let us clarify your final concerns.
> > >
> > > 1. We understand your concern. Let us first recall that all the proofs of the propositions presented in the manuscript are provided in Appendix C. Concerning the correctness, observe that the proposed algorithm is simply an alternating minimization scheme, for which the convergence to a stationary point in the non-convex setting has been extensively studied in the literature (e.g. [1]). We agree that in practice, we cannot exactly solve (3) (or equivalently (5)), and we rely on the approximate solution obtained by GPTQ. However, one of the major practical interest of LRC, as Reviewer **TKaQ** aptly noted, is that our `method can use any weight quantization technique as a subroutine (they use GPTQ in the paper) `, and incorporating the errors induced by GPTQ into the convergence analysis of our scheme would undermine the fundamental objectives of the approach. **Therefore we believe your impression that the paper "lacks of theoretical rigor" should be balanced with the proofs derived, and the correctness of the algorithm (assuming exact resolution of (5)). Additionally, we assert that the main interest of our approach lies in its practical applications, as demonstrated by the extensive experiments we have conducted.**
> > >
> > > 2. We hear you when you say that you feel our work is limited in terms of novelty. We agree that our work is a closely related to some prior works (e.g. [2, 3, 4]) , as they all leverage low-rank weights to improve the quantization process. However, these works mostly focus on the quantization of the weights and completely discard the quantization of activations, which is the main motivation of why one should add low-rank weights in the first place. In the weight only quantization setting, several other approaches have already closed the gap with the full precision model (see Table 3), and therefore the introduction of low-rank correction terms is not needed.  Our work shows that low-rank correction terms significantly reduce the accuracy gap with original models when activations are also quantized. **This substile difference positions our work as a novel quantization scheme, primarily aimed at mitigating quantization errors in activations.**
> > >
> > >
> > > **We hope these explanations address your concerns** and we sincerely appreciate the time you have put into reading our rebuttal.
> > >
> > > [1] Beck, Amir "On the convergence of alternating minimization for convex programming with applications to iteratively reweighted least squares and decomposition schemes", SIAM Journal on Optimization, 2015
> > >
> > > [2]  Ou, Lin, et al. "Adaptive quantization error reconstruction for llms with mixed precision", First Conference on Language Modeling, 2024
> > >
> > > [3] Rajarshi, Saha, et al. "Compressing large language models using low rank and low precision decomposition", arXiv preprint
> > > arXiv:2405.18886, 2024
> > >
> > > [4] Zhang, Cheng, et al. "Lqer: Low-rank quantization error reconstruction for llms", Forty-first International Conference on Machine Learning, 2024

---

> > > > ### Author Response · Authors · 2024-11-30
> > > > **Final Concerns**
> > > >
> > > > Dear Reviewer,
> > > >
> > > > Thank you once again for reviewing our rebuttal and for your prompt response. We would be more than happy to address any remaining concerns you may have.
> > > >
> > > > Please let us know if there are any specific points in our work that you feel need further improvement. We would be grateful for your additional suggestions and are eager to make the necessary revisions.
> > > >
> > > > Thank you again for your review and your time.

---

### Official Review · Reviewer_xtpe · 2024-11-04

**Soundness:** 2
**Presentation:** 3
**Contribution:** 3
**Rating:** 5
**Confidence:** 3

**Summary:**

The paper studies post training quantization. They achieve a new SoTA for W4A4. This and the introduction of their algorithm is the main contribution of the paper. The main idea is to solve Equation (2) using alternating minimization by optimizing either for quantized weights or for low rank matrices. The authors also initialize the low rank matrices carefully. In terms of experimental results, the authors report wikitext-2 perplexity as well as from lm-eval (PIQA, HellaSwag, Arc-Easy, Arc-Challenge, Winograd and Lambada). The numbers demonstrate that the introduced approach beat their main benchmark QuaRot. Surprisingly, one iteration of alternating minimization is enough to achieve good performance.

**Strengths:**

please see summary

**Weaknesses:**

minor typos:

Line 57: summurized -> summarized

Line 191: "... we drop the dependency in l of our notations" -> replace "in" with "on"

Line 353: "Our ambition is to close the gap between out main benchmark" -> replace "out" with "our"?

Line 405: "equan" -> equal

Line 421: "prononced" -> pronounced

Line 455: "In this work we have not study the computational" -> replace "study" with "studied"

**Questions:**

The paper contains several propositions but they lack proofs.

The approach of adding a low rank matrix is similar to approaches in these papers that the authors could cite: "LoRA: Low-Rank Adaptation of Large Language Models" and "QLoRA: Efficient Finetuning of Quantized LLMs".

My main concern with the paper is that it doesn't cite or discuss the above 2 related papers, which seem quite related. Also, the paper lacks several proofs. Furthermore, the paper could consider generative tasks for benchmarks such as gsm8k. Typically, generative tasks are harder for quantized model than multiple choice tasks. Given all this, my overall rating for the paper is "marginally below the acceptance threshold".

---

> ### Author Response · Authors · 2024-11-22
> **Rebuttal by Authors**
>
> Thank you for poiting out the typos. We have corrected them.
>
> >The paper contains several propositions but they lack proofs.
>
> Thank you for poiting out this omission. We have added all the proofs in the Appendix C.
>
> >The approach of adding a low rank matrix is similar to approaches in these papers that the authors could cite: "LoRA: Low-Rank Adaptation of Large Language Models" and "QLoRA: Efficient Finetuning of Quantized LLMs".
>
> These papers aims to tackle the finetuning stage of LLMs rather than their post-training quantizations, but we agree that they rely on additional low-rank weights. We have followed the reviewer suggestion and cited them in the Introduction.
>
> >Furthermore, the paper could consider generative tasks for benchmarks such as gsm8k. Typically, generative tasks are harder for quantized model than multiple choice tasks.
>
> We agree that we did not evaluate on generative tasks. However, we are relying on the lm-eval datasets that are currently the standard evaluation benchmarks used to measure the performances of quantized models. Please refer to this very recent paper (Spotlight NeurIPS 2024) as an example of such an evaluation "QTIP: Quantization with Trellises and Incoherence Processing". Additionally, we have reported the perplexities (PPL) obtained by the various approaches in all our tables for a comprehensive evaluation of their performances.

---

> ### Author Response · Authors · 2024-11-30
> **Kind Reminder**
>
> Dear Reviewer,
>
> Thank you for your comments and review. We believe that we have addressed all your concerns and have implemented the necessary changes in the current version of the manuscript. Additionally, we noticed that you rated our manuscript with a score of **1** for **Presentation**, which appears to be an outlier compared to the comments of other reviewers. We understand that this decision was based on your initial feedback, but we hope that the issues you raised have now been resolved. If you agree with our assessment, we kindly ask you to reconsider your overall score.
>
> Thank you again for your time and your review.

---

> > ### Comment · Reviewer_xtpe · 2024-12-03
> > **Reviewer response**
> >
> > We thank the authors for addressing our concerns. I'm willing to increase the presentation score to 3

---

> ### Author Response · Authors · 2024-12-03
> **Thank you for your response**
>
> Dear Reviewer,
>
> We sincerely appreciate your thoughtful review and are delighted to hear that we successfully addressed all your concerns. We are also grateful for the increase of **2 points** in the **Presentation** score.
>
> While the detailed scores have seen significant improvement and all concerns have been resolved, we noticed that the overall score remains unchanged. We kindly request a reconsideration of the overall rating to better reflect the reevaluation of the manuscript.
>
> Thank you once again for your time and effort in reviewing our work.

---

### Official Review · Reviewer_RmtB · 2024-11-04

**Soundness:** 2
**Presentation:** 3
**Contribution:** 2
**Rating:** 5
**Confidence:** 2

**Summary:**

This paper introduces a new method called LRC (Low-Rank Correction) for quantizing large language models (LLMs) to 4-bit weights and activations (W4A4) while minimizing accuracy loss. The key idea is to jointly optimize for a quantized weight matrix acting on quantized activations and a full-precision low-rank weight matrix acting on the original unquantized activations. This allows LRC to significantly reduce the quantization error and close the accuracy gap with the full-precision model.

**Strengths:**

This paper is its novel approach to quantizing large language models to 4-bit weights and activations while maintaining high accuracy. The LRC method's ability to optimize jointly for a quantized weight matrix and a full-precision low-rank correction matrix, which is connected to the original unquantized activations, effectively reduces quantization error. This innovative technique sets LRC apart from previous approaches and demonstrates its potential for enabling highly compressed models with minimal performance degradation.

**Weaknesses:**

The paper does not analyze the computational cost associated with the added low-rank correction matrix. While the method effectively reduces quantization error, the impact on inference time and memory usage is not thoroughly explored. This is an important consideration for the practical deployment of the LRC method.

The authors leave the ideal implementation of the low-rank computation for future work. Without a concrete implementation strategy, it may be difficult for practitioners to immediately adopt the LRC method in real-world applications. Providing guidance or preliminary implementation details could have made the paper more impactful.

Although the paper identifies activation quantization as the primary source of error, it does not propose novel activation quantization schemes. The authors rely on existing techniques like round-to-nearest and suggest that future work on improved activation quantization could lead to better results. Addressing this limitation within the paper could have further strengthened the contribution.

**Questions:**

How does the computational cost of the low-rank correction matrix scale with the size of the language model? Is there a trade-off between the compression ratio and the computational overhead introduced by LRC?

Can the LRC method be extended to other model compression techniques, such as pruning or knowledge distillation? How would the low-rank correction approach interact with these techniques?

How sensitive is the performance of LRC to the choice of calibration dataset used for computing the activation statistics? Would using a more diverse or domain-specific calibration dataset lead to better results?

The authors suggest that improved activation quantization schemes could further enhance the performance of LRC. What specific properties should these improved schemes have, and how might they be developed?

---

> ### Author Response · Authors · 2024-11-22
> **Rebuttal by Authors: Part 1/2**
>
> >The paper does not analyze the computational cost associated with the added low-rank correction matrix. While the method effectively reduces quantization error, the impact on inference time and memory usage is not thoroughly explored.
>
> We thank the reviewer for this important remark. Currently we are only emulating the implementation of our approach and do not leverage a specific cuda kernel adapted to our computational scheme in order to perform the low-rank correction. However, for the sake of evaluation, we have added an experiment evaluating the speedup obtained with a naive cuda kernel implementing our approach and compare it against the full precision version. Additionally, we have added the sizes of the different models when the rank is set to 10% of the original sizes. Both experiments are presented at the beginning of the rebuttal in the paragraphs titled 'Experiment: Latency of LRC' and 'Experiment: Memory Footprint of LRC', respectively.
>
>
> >The authors leave the ideal implementation of the low-rank computation for future work [...]. Providing guidance or preliminary implementation details could have made the paper more impactful.
>
> Thank you for this insightful comment. We followed the reviewer's suggestion and included in Appendix B.2. the discussion from our new experiment on latency, as presented at the beginning of the rebuttal. Additionally, we will clarify that the proposed kernel is a naive implementation of LRC that performs operations sequentially. We hypothesize that these operations can be executed in parallel.
>
> >Although the paper identifies activation quantization as the primary source of error, it does not propose novel activation quantization schemes [...]. Addressing this limitation within the paper could have further strengthened the contribution.
>
> Thank you for your comment. We would like to highlight that LRC is a novel quantization scheme primarily designed to mitigate quantization errors in _activations_. Notably, when activations remain unquantized, the introduction of additional low-rank weight matrices exerts minimal impact on the performance (see Table 3), and low-rank correction terms are not needed in this setting. Therefore, although LRC is structured as additional weights, its principal objective is to rectify the errors associated with activation quantization. We acknowledge that LRC is currently presented using RTN for activation quantization (and GPTQ for weight quantization), however the framework proposed allows other quantization techniques to be applied instead. We believe that improving RTN is definitively an opportunity for further enhancement, but we consider it beyond the scope of this work.
>
>
> >Is there a trade-off between the compression ratio and the computational overhead introduced by LRC?
>
> In an idealized scenario, it would be possible to design a CUDA kernel that computes both the full and low-rank terms in parallel, thereby eliminating any computational tradeoff for LRC. However, such a kernel is not currently available, resulting in a tradeoff between accuracy and computational overhead. While we did not investigate this specific tradeoff, Figure 2 (and Figure 4 in Appendix B.3) illustrates a similar tradeoff by measuring accuracy against the chosen rank.
>
> >Can the LRC method be extended to other model compression techniques, such as pruning or knowledge distillation? How would the low-rank correction approach interact with these techniques?
>
> That is an excellent point, thank you. For instance, in the context of pruning, if one aims to correct the pruned model by incorporating additional low-rank weights, it would be feasible to substitute Eq. (5) with a layer-wise pruning objective while employing the same computational scheme as the one proposed in this work (LRC). Although this represents an intriguing application of our work, we consider it beyond the scope of this study.

---

> ### Author Response · Authors · 2024-11-27
> **Rebuttal by Authors: Part 2/2**
>
> >How sensitive is the performance of LRC to the choice of calibration dataset used for computing the activation statistics? Would using a more diverse or domain-specific calibration dataset lead to better results?
>
> Thank you for this suggestion. We have performed an additional experiment showing the effect of the choice of the calibration dataset on the performances of LRC. Please refer to the experiment presented at the beginning of the rebuttal in the paragraph titled "Experiment: Effect of the Calibration Dataset" to see the results obtained. We have also added these results in Appendix B.1.
>
> >The authors suggest that improved activation quantization schemes could further enhance the performance of LRC. What specific properties should these improved schemes have, and how might they be developed?
>
> Although LRC does not impose specific requirements on the quantization scheme for activations, we consider this step to be a fundamental limitation of current quantization techniques. Currently LRC can mitigate this gap by incorporating low-rank terms, but this may not suffice to fully recover the original model performance if a sufficiently small rank is chosen. Therefore, to enhance the efficiency of LRC, it might be essential to improve the quantization of activations.

---

> ### Author Response · Authors · 2024-11-30
> **Kind Reminder**
>
> Dear Reviewer,
>
> We sincerely appreciate your valuable comments and questions, which have greatly contributed to improving our manuscript. We believe that we have thoroughly addressed all your concerns and appreciate the opportunity to clarify our work. As the rebuttal period draws to a close, we would like to confirm whether our responses have adequately addressed your concerns.
>
> Thank you again for your time and your review.

---

### Official Review · Reviewer_iJC7 · 2024-11-04

**Soundness:** 3
**Presentation:** 3
**Contribution:** 2
**Rating:** 6
**Confidence:** 2

**Summary:**

This work studies a brand-new approach on the post-training quantization problem of LLMs. In particular, the authors suggest introducing a low-rank adaptation to relieve the accuracy loss of quantization, which uses a small size dataset to generalize their low-rank adaptation matrix U, V. This work is novel and easy to implement.

**Strengths:**

* The idea of introducing low-rank adaptation to correct the quantization error is good, and trivially effective. In my perspective, these adaptation-based methods are worthy of further and comprehensive study.
* The derivation in this paper provides concise intuition, which is easy to follow.
* The topic of efficient LLM deployment is becoming vital currently, this method has considerable potential in addressing such PTQ problems on LLMs.

**Weaknesses:**

* It is not clear how the rank of adaptation would influence the efficiency. This would become my main concern for this paper. I strongly recommend the authors to add an experiment to evaluate its enhancement of memory usage and speed.
* The presentation of this paper is good, but not excellent enough. The authors should add an introduction of GPTQ method and Cholesky in their appendix (since they are parts of the main algorithm) for presenting to the broader audience.
* The choice of dataset in $X$ should be carefully considered, but I don’t see any analysis about how $X$ will affect the performance. I recommend the authors to add a discussion about this and their assumption for rigorousness.

**Questions:**

* Please answer the issues and questions in the Weakness and point out my potential misunderstandings. I am happy to discuss and enhance my rate.
* Why is the last line (LRC (5)) for the Mistral model missing both in Table 1 and Table 5?
* From Table 3, LRC fails to beat SVD on average score, can you explain why this happens?

---

> ### Author Response · Authors · 2024-11-22
> **Rebuttal by Authors: Part 1/2**
>
> >It is not clear how the rank of adaptation would influence the efficiency. This would become my main concern for this paper. I strongly recommend the authors to add an experiment to evaluate its enhancement of memory usage and speed.
>
> We thank the reviewer for this important remark. Currently we are only emulating the implementation of our approach and do not leverage a specific cuda kernel adapted to our computational scheme in order to perform the low-rank correction. However, for the sake of evaluation, we have added an experiment evaluating the speedup obtained with a naive cuda kernel implementing our approach and compare it against the full precision version. Additionally, we have added the sizes of the different models when the rank is set to 10% of the original sizes. Both experiments are presented at the beginning of the rebuttal in the paragraphs titled 'Experiment: Latency of LRC' and 'Experiment: Memory Footprint of LRC', respectively.
>
>
> >The presentation of this paper is good, but not excellent enough. The authors should add an introduction of GPTQ method and Cholesky in their appendix (since they are parts of the main algorithm) for presenting to the broader audience.
>
> Thank you for acknowledging the quality of our presentation. We have followed the suggestion of the reviewer and added additional background on Cholesky and GPTQ in Appendix A. For the convenience of the reader, we also report the added paragraphs below.
>
> **GPTQ Algorithm.** The GPTQ algorithm, introduced by Frantar et al. (2022), is a post-training quantization technique designed to efficiently reduce the precision of weights in large language models (LLMs) while maintaining their performance. To achieve this, the authors propose to approximate a solution of the layer-wise quadratic approximation problem defined as:
>
> $$\min_{\widehat{\textbf{W}}\in\mathcal{C}(b)\cap \mathbb{R}^{d^{\text{out}}\times d^{\text{in}}}}\mathcal{L}_{\text{q}}(\widehat{\textbf{W}}):=\Vert \textbf{W}\textbf{X} - \widehat{\textbf{W}} \textbf{X}\Vert_2^2\; $$
>
> where $\textbf{W}$ is the original weight matrix, and $\mathcal{C}(b)$ is the constraint set of matrices admitting a certain bit per weight $b>0$ precision. The main difficulty of solving exactly this optimization problem resides in the constraint set $\mathcal{C}(b)$, making the problem non-convex. To approximate a solution, Frantar et al. (2022) propose to improve the computational scheme of the greedy approach originally proposed by LeCun et al. (1989) for pruning, and then adapted for quantization in  (Frantar & Alistarh,
> 2022), by removing the ordering in the greedy quantization process, and applying the algorithm in parallel over multiple columns.
>
>
>
> **Cholesky Factorization.** Cholesky factorization is a numerical method used to decompose a symmetric positive-definite matrix (PD) into the product of a lower triangular matrix with positive diagonal coefficients and its transpose. This technique is particularly useful in solving systems of linear equations, performing matrix inversion, and computing the determinant of a matrix. More formally given $\Sigma$ a symmetric PD matrix, there exists a unique lower triangular matrix $\textbf{L}$ such that
>
> $$\Sigma = \textbf{L}\textbf{L}^\top .$$
>
> To compute the Cholesky factor $\textbf{L}$, one can rely on the Cholesky Algorithm which is a modified version of the Gaussian elimination and requires $\mathcal{O}(n^3)$ FLOPs where $n$ is the size of $\Sigma$.
>
> >The choice of dataset in $X$ should be carefully considered, but I don’t see any analysis about how $X$ will affect the performance. I recommend the authors to add a discussion about this and their assumption for rigorousness.
>
> Thank you for this suggestion. We have performed an additional experiment showing the effect of the choice of the calibration dataset on the performances of LRC. Please refer to the experiment presented at the beginning of the rebuttal in the paragraph titled "Experiment: Effect of the Calibration Dataset" to see the results obtained. We have also added these results in Appendix B.1.
>
> >Please answer the issues and questions in the Weakness and point out my potential misunderstandings. I am happy to discuss and enhance my rate.
>
> Thank you. We hope we have answered your questions, and we would be delighted to continue the discussion if any points require further explanation.

---

> > ### Author Response · Authors · 2024-11-27
> > **Rebuttal by Authors: Part 2/2**
> >
> > >Why is the last line (LRC (5)) for the Mistral model missing both in Table 1 and Table 5?
> >
> > Thank you for spotting this. We have corrected the manuscript. Please find below the missing lines for both tables:
> >
> > Table 1:
> > | Method       | Model   | PPL  | PQ    | HS    | A-e   | A-c   | WG    | LA    | Avg  |
> > |--------------|---------|------|-------|-------|-------|-------|-------|-------|------|
> > | `LRC (5)`    | Phi-3   | 7.2  | 0.77  | 0.734 | 0.799 | 0.545 | 0.668 | 0.639 | 0.693|
> > | `LRC (5)`    | Llama-3  | 7.94 | 0.764 | 0.742 | 0.758 | 0.483 | 0.705 | 0.739 | 0.698|
> > | `LRC (5)`    | Mixtral | 4.41 | 0.801 | 0.8   | 0.813 | 0.555 | 0.736 | 0.814 | 0.753|
> >
> >
> > Table 2:
> > | Method       | Model   | PPL  | PQ    | HS    | A-e   | A-c   | WG    | LA    | Avg  |
> > |--------------|---------|------|-------|-------|-------|-------|-------|-------|------|
> > | `LRC (5)`    | Phi-3   | 7.25 | 0.776 | 0.728 | 0.797 | 0.539 | 0.706 | 0.65  | 0.699|
> > | `LRC (5)`    |  Llama-3  | 7.02 | 0.783 | 0.761 | 0.766 | 0.494 | 0.735 | 0.765 | 0.717|
> > | `LRC (5)`    | Mixtral | 4.25 | 0.817 | 0.812 | 0.817 | 0.572 | 0.738 | 0.815 | 0.762|
> >
> >
> >
> > >From Table 3, LRC fails to beat SVD on average score, can you explain why this happens?
> >
> > When only the weights are quantized, as shown in Table 3, methods incorporating additional low-rank weight matrices (e.g., LRC and SVD) exhibit performance comparable to the basic QuaRot model and even the original model. In this context, low-rank terms have minimal impact, as the baseline performance of QuaRot is nearly lossless, as we explained l. 383 of the manuscript.

---

> > > ### Comment · Reviewer_iJC7 · 2024-11-28
> > >
> > > Thank you for your rebuttal. In a word, the rebuttal has already solved most of my concerns. I thus will enhance my rating score from 5 to 6.
> > >
> > > However, this paper still needs some polish for the camera-ready version or the next submission. I adjust my confidence from 3 to 2 since there exist some issues the author can improve to make the manuscript better. I list them in the following:
> > >
> > > * The methodology part takes 4 pages in the manuscript, whereas the novelty of this method is not a top-level one, so the contribution of this paper will be limited since the authors already spent 4 pages on introducing their approach. I suggest compressing this part as concisely as possible to provide clear and straight intuition.
> > >
> > > * For the rest of the first 10 pages. A clear claim to your goal and the problem you want to solve in this paper will greatly help clarify this work's contribution. Besides, doing more experiments on more models and tasks can also improve the contribution, if possible, make at least over 50 % of the first 10 pages of the experimental part.
> > >
> > > * Supplementing the complete error bar for all results and adding an ablation study will be also helpful.
> > >
> > > Conclusively, you should provide a perfect experimental part to demonstrate the impact of this work while making this method as easy as possible to be implemented. Best of luck to you.

---

> > > > ### Author Response · Authors · 2024-11-28
> > > > **Many thanks for reading our rebuttal**
> > > >
> > > > Dear Reviewer,
> > > >
> > > > Many thanks for reacting to our rebuttal. We are pleased to note that we have addressed most of your concerns and deeply appreciate your decision to revise your score, despite the remaining issues.
> > > >
> > > > We are also grateful for your constructive feedback. In response, we have refined the presentation of our methodology to make it more concise and have clearly articulated the objectives of our study throughout the paper, as well as at the beginning of the experimental section. Furthermore, we have expanded our empirical study by incorporating two additional benchmarks with Llama-2 (7B and 13B). Finally, to provide a more comprehensive analysis of the rank's effect in LRC, we have included results for Llama-3 (detailed in Appendix C.3) and Mixtral (see the updated Figure 2). Please refer to the updated version of the manuscript for these changes. We will also follow your suggestion and add error bars for all results in the final version.
> > > >
> > > > We hope that these clarifications adequately address your final concerns.
> > > >
> > > > Thanks again for reading our rebuttal, and for your detailed review.

---

### Official Review · Reviewer_TKaQ · 2024-11-05

**Soundness:** 3
**Presentation:** 3
**Contribution:** 2
**Rating:** 6
**Confidence:** 3

**Summary:**

The paper proposes LRC (Low Rank Correction), a method to perform error correction for quantization of weights *and* activations (a common approach to perform model compression in LLMs). They add a full precision low-rank weight matrices in the forward pass that act on the unquantized activations and accounts for errors arising from both weights and activations.

**Strengths:**

- They paper is written well and easy to understand. The scheme they propose is sound and intuitive in its formulation.
- Their method can use any weight quantization technique as a subroutine (they use GPTQ in the paper), which allows other tools/papers to plugin their own method.
- They perform sensible ablations in order to clearly identify the impact of the weight only quantization vs activation quantization, and when low rank error correction offers value. Moreover, they also show that the quantization is replaceable by any technique which can use the weight matrix and the covariances matrices to output a quantized version.

**Weaknesses:**

# Major
- *Limited Contribution*: The paper stitches together many well known building blocks in the PTQ literature to build a sane, effective technique. In my opinion, it is a sound engineering feat, but still has high overlap with the previous work on the topic by Zhang et al (2024) and Ou et al (2024). The authors do differentiate themselves by the fact that they do a joint optimization over the low rank and quantized matrices which is key to the delta over the previous work. However, this is also a well known technique and has been applied in other works such as [1]
- The experiments are performed on models which do not overlap with the QuaRot paper (their primary baseline) and hence it is difficult to compare directly. Moreover, their method uses additional bits (aka low rank correction factors) and hence comparison to QuaRot is not a strictly fair comparison, a small increase in model size though it may be.
- The authors mention a number of contemporary work (Quip, Quip#, LQER etc) but provide an empirical comparison against a very limited baseline (QuaRot).
- The authors claim that they can close the accuracy gap using rank equivalent to 30%, this seems to be true only for Phi3 which the authors present as an unqualified statement. Moreover, I cannot find the corresponding results in the evaluation section

# Minor
- In L431, the authors use the phrase "10% additional ranks" which I believe is not meaningful
- L399-407 sub section "on the effect of rank" does not mention the table/fig where the results can be found
- Table 1, for Phi3, LRC(1) outperforms LRC(5), a weird anomaly since one would expect performance to increase monotonically with iters
- Best results not provided in bold font
# References
1. Saha, Rajarshi, et al. "Compressing Large Language Models using Low Rank and Low Precision Decomposition." arXiv preprint arXiv:2405.18886 (2024).

**Questions:**

- Why is it necessary to store the low rank matrices in full precision? Couldn't they also be quantized?

---

> ### Author Response · Authors · 2024-11-22
> **Rebuttal by Authors: Part 1/3**
>
> >Limited Contribution: The paper stitches together many well known building blocks in the PTQ literature to build a sane, effective technique. In my opinion, it is a sound engineering feat, but still has high overlap with the previous work on the topic by Zhang et al (2024) and Ou et al (2024).
>
> We agree that our work is a closely related to the prior works cited, as they all leverage low-rank weights to improve the quantization process. Note however that Ou et al. (2024) is improving on Zhang et al (2024) by proposing to replace the SVD factorization by a projection which takes into account the statistical properties of the output activations, however they completely discard the quantization of activations, which is in our opinion, the main motiviation of why one should add low-rank weights in the first place. In this work, not only we improve the proposed factorization of Ou et al.(2024) (and as a direct consequence the one proposed in Zhang et al (2024)) but most importantly we incorporate the effect of quantizing activations into the quantization process. The latter being the main reason why one would require addditional low-rank weight matrices to correct quantization errors.
>
> >This is also a well known technique and has been applied in other works such as [1]
>
> Thank you for highlighting this reference. We have included the following discussion of this work in our manuscript:
>
> In [1], the authors improve the methodology of Ou et al.(2024) by considering a joint formulation of the quantization problem to optimize for both the quantized weights and the low-rank terms. However, the authors only focus on the quantization of the weights, leaving aside the quantization of activations. In this work, we also consider a joint formulation, however our focus is on improving the quantization of activations. We improve on prior research by incorporating both the empirical distribution of activations and the errors induced by activation quantization into our analysis to optimize the low-rank weight matrices.
>
> We would like also to mention that for weight only quantization, several other approaches have already closed the gap with the full precision model (see Table 3), and therefore the introduction of low-rank correction terms might not be needed. Our work shows that low-rank correction terms significantly reduce the accuracy gap with original models when activations are also quantized.

---

> ### Author Response · Authors · 2024-11-22
> **Rebuttal by Authors: Part 2/3**
>
> >The experiments are performed on models which do not overlap with the QuaRot paper (their primary baseline) and hence it is difficult to compare directly.
>
> Concerning this point, we thought it would be more interesting to compare our method with Quarot's performance on more recent LLMs. We did replicate the QuaRot method (using code from those authors), but we are delighted to compare our method with Quarot on the LLMs used in the original paper. We have added to the tables 1 and 2 of the manuscript the performances obtained by the different quantization methods on Llama 2 7B and 13B.  For the convenience of the reader, we state the results again here:
>
>
> Llama 2 (7B) without groupsizing
> | Method       | PPL  | PQ    | HS    | A-e   | A-c   | WG    | LA    | Avg.  |
> |--------------|------|-------|-------|-------|-------|-------|-------|-------|
> | FP16         | 5.47 | 0.791 | 0.76  | 0.745 | 0.462 | 0.691 | 0.739 | 0.698 |
> | QuaRot       | 6.13 | 0.77  | 0.728 | 0.703 | 0.417 | 0.663 | 0.712 | 0.665 |
> | SVD          | 6.12 | 0.77  | 0.729 | 0.711 | 0.436 | 0.665 | 0.717 | 0.671 |
> | LRC (1)      | 5.77 | **0.776** | 0.731 | 0.726 | 0.424 | **0.676** | 0.747 | 0.68  |
> | LRC (5)      | **5.75** | 0.774 | **0.733** | **0.727** | **0.439** | 0.669 | **0.748** | **0.682** |
>
> Llama 2 (7B) with groupsizing (128)
> | Method       | PPL  | PQ    | HS    | A-e   | A-c   | WG    | LA    | Avg.  |
> |--------------|------|-------|-------|-------|-------|-------|-------|-------|
> | FP16 | 5.47 | 0.791 | 0.76  | 0.745 | 0.462 | 0.691 | 0.739 | 0.698 |
> | QuaRot | 6.12 | 0.763 | 0.725 | 0.701 | 0.41  | 0.669 | 0.715 | 0.664 |
> | SVD   | 6.11 | 0.778 | 0.725 | 0.694 | 0.416 | 0.657 | 0.718 | 0.665 |
> | LRC (1) | 5.69 | 0.779 | **0.734** | **0.736** | **0.444** | 0.672 | 0.748 | **0.685** |
> | LRC (5) | **5.68** | **0.78** | **0.734** | 0.727 | 0.434 | **0.677** | **0.747** | 0.683 |
>
>
> Llama 2 (13B) without groupsizing
> | Method       | PPL  | PQ    | HS    | A-e   | A-c   | WG    | LA    | Avg.  |
> |--------------|------|-------|-------|-------|-------|-------|-------|-------|
> | FP16         | 4.88 | 0.805 | 0.794 | 0.774 | 0.491 | 0.721 | 0.767 | 0.725 |
> | QuaRot       | 5.34 | 0.784 | 0.767 | 0.755 | 0.481 | 0.709 | 0.747 | 0.707 |
> | SVD          | 5.31 | 0.792 | 0.772 | 0.755 | **0.486** | 0.699 | 0.747 | 0.709 |
> | LRC (1)      | 5.09 | **0.788** | 0.77  | 0.764 | 0.482 | 0.702 | **0.781** | 0.715 |
> | LRC (5)      | **5.08** | 0.786 | **0.774** | **0.769** | 0.478 | **0.706** | **0.781** | **0.716** |
>
>
> Llama 2 (13B) with groupsizing (128)
> | Method       | PPL  | PQ    | HS    | A-e   | A-c   | WG    | LA    | Avg.  |
> |--------------|------|-------|-------|-------|-------|-------|-------|-------|
> | FP16 | 4.88 | 0.805 | 0.794 | 0.774 | 0.491 | 0.721 | 0.767 | 0.725 |
> | QuaRot | 5.35 | 0.782 | 0.762 | 0.758 | 0.472 | 0.702 | 0.75  | 0.705 |
> | SVD   | 5.34 | 0.783 | 0.768 | 0.748 | 0.476 | 0.699 | 0.753 | 0.705 |
> | LRC (1) | 5.05 | 0.789 | **0.777** | **0.763** | **0.491** | **0.717** | **0.783** | **0.72** |
> | LRC (5) | **5.04** | **0.798** | 0.776 | 0.762 | **0.491** | 0.7   | 0.78  | 0.718 |
>
> >Moreover, their method uses additional bits (aka low rank correction factors) and hence comparison to QuaRot is not a strictly fair comparison, a small increase in model size though it may be.
>
> We agree that LRC incurs additional memory footprint. More precisely, in our experiments, we settled on adding low-rank weight matrices with a rank corresponding to 10% of the original size, therefore we are effectively at 5.6 bits (4 + 0.1 * 16) per weight. We argue that this additional memory usage is worth spending for improved downstream-task performance. Note also that the SVD approach (LQER) requires the exact same bit precision. We will clarify this point in the manuscript.
>
> >The authors mention a number of contemporary work (Quip, Quip#, LQER etc) but provide an empirical comparison against a very limited baseline (QuaRot).
>
> We focus mainly on comparing with QuaRot as it was the SoTA approach (at the time of the submission) to quantize LLMs in the weights-and-activations setting. Note also that Quip and Quip# do not handle the case where both weights and activations are quantized. Finally, we do compare with an improved version of LQER (where QuaRot is additionally applied) which we called SVD in tables 1,2 and 3.

---

> ### Author Response · Authors · 2024-11-23
> **Rebuttal by Authors: Part 3/3**
>
> >The authors claim that they can close the accuracy gap using rank equivalent to 30%, this seems to be true only for Phi3 which the authors present as an unqualified statement. Moreover, I cannot find the corresponding results in the evaluation section.
>
> Thanks for pointing this. We have conducted additional experiments showing that using a rank of 30% of the original sizes at W4A4 enables to consistently recover the performances of the full precision model.  Please find below the comparison between the full precision models and their quantized versions with LRC at W4A4 and a rank set at 30\%. More details can also be found in Appendix B.3.
>
> | Method       | Model   | acc_avg | arc_challenge | arc_easy | hellaswag | lambada_openai | piqa  | winogrande |
> |--------------|---------|---------|---------------|----------|-----------|----------------|-------|------------|
> | FP16         | Llama-3 | 0.7328  | 0.5333        | 0.7778   | 0.7916    | 0.7605         | 0.8074| 0.7261     |
> | LRC     | Llama-3 | 0.7284  | 0.5171        | 0.7879   | 0.7749    | 0.7722         | 0.7938| 0.7245     |
>
>
> | Method       | Model   | acc_avg | arc_challenge | arc_easy | hellaswag | lambada_openai | piqa  | winogrande |
> |--------------|---------|---------|---------------|----------|-----------|----------------|-------|------------|
> | FP16         | Phi-3   | 0.7203  | 0.5657        | 0.7858   | 0.7749    | 0.6534         | 0.8085| 0.7332     |
> | LRC     | Phi-3   | 0.7185  | 0.5742        | 0.8005   | 0.7523    | 0.6577         | 0.796 | 0.7301     |
>
>
> | Method       | Model   | acc_avg | arc_challenge | arc_easy | hellaswag | lambada_openai | piqa  | winogrande |
> |--------------|---------|---------|---------------|----------|-----------|----------------|-------|------------|
> | FP16         | Mixtral | 0.7762  | 0.5964        | 0.8338   | 0.8399    | 0.7842         | 0.8373| 0.7656     |
> | LRC     | Mixtral | 0.7767  | 0.5964        | 0.8346   | 0.826x     | 0.811          | 0.8308| 0.7616     |
>
>
> >In L431, the authors use the phrase "10% additional ranks" which I believe is not meaningful.
>
> We have corrected this formulation by "by incorporating low-rank weight matrices with ranks set to 10\% of the original matrix sizes".
>
> >L399-407 sub section "on the effect of rank" does not mention the table/fig where the results can be found.
>
> Thank you. We have corrected this error and referred to Figure 2.
>
> >Table 1, for Phi3, LRC(1) outperforms LRC(5), a weird anomaly since one would expect performance to increase monotonically with iters
>
> Thank you for pointing this out. This phenomenon may arise from numerical instabilities associated with matrix decomposition and inversion processes inherent in LRC. To mitigate this undesirable effect, we are conducting experiments to optimize the regularization parameters $\epsilon_x$ and $\epsilon_y$ as specified in line 303. We anticipate presenting conclusive results from these experiments in the coming days.
>
> >Best results not provided in bold font
>
> We have corrected this, thank you.
>
>
> >Why is it necessary to store the low rank matrices in full precision? Couldn't they also be quantized?
>
> Certainly, it is indeed feasible to quantize these matrices, which would result in a small reduction in memory footprint. However, it is unlikely that this would lead to significant improvements in throughput. This is because the low-rank matrix multiplications are memory-bound, and in this scenario, the activations **X**  (which are _not_ quantized) will dominate the throughput. To address this, we would need a sequence of operations that compute low-rank matrix multiplications in mixed precision (for the non-quantized low-rank weight matrices and the activations), followed by rescaling for each low-rank matrix. Fusing these operations is challenging with the current PyTorch tools, although a dedicated kernel could potentially achieve good throughput. We leave these details for future work

---

> > ### Comment · Reviewer_TKaQ · 2024-11-26
> > **Comprehensive encouraging empirical results**
> >
> > Thank you for all your effort and the detailed rebuttal.
> >
> > # Part 1/3 (Contribution)
> >
> > I agree that the quantization of activations is less explored compared to weight quantization techniques, and the application of low rank matrices is only relevant to activation quantization (based on Table 3)
> >
> > # Part 2/3 (Comparison w/QuaRot)
> >
> > Thank you for reproducing the results here for both the newer models and the models present in the original QuaRot paper. While I am convinced based on the results that the method outperforms QuaRot (which is also expected given that it performs more work and uses more memory than QuaRot). My intent is not to dispute whether memory is worth spending but that the increase in memory should be explicitly stated in a table column (similar to average bit precision or bits per weight in Ou et al) so that the reader can make an informed choice.
> >
> > # Part 3/3 (More experiments)
> >
> > Why does LRC outperform FP16 on so many cases? lambada_openai for all models, ARC-C and ARC-E for mixtral and Phi-3 etc
> > It does appear that 30% is good enough to close the gap in most cases, and sensible starting point for tuning the rank of the correction factors.
> >
> > Based on all the other comments and the rebuttal, I am willing to raise my score.

---

> ### Author Response · Authors · 2024-11-27
> **Many thanks for reading our rebuttal**
>
> Dear Reviewer,
>
> We sincerely appreciate your valuable comments and questions, which have greatly contributed to improving our manuscript. We are also grateful for the increase in your score.
>
> > My intent is not to dispute whether memory is worth spending but that the increase in memory should be explicitly stated in a table column (similar to average bit precision or bits per weight in Ou et al) so that the reader can make an informed choice.
>
> Thank you for clarifying this point. We definitively agree that the reader should be informed about the memory footprint required by LRC, and we have added a column in Table 3 of the manuscript that reports the model size obtained by the different quantization methods. These results are also detailed at the beginning of the rebuttal in the paragraph titled "Experiment: Memory Footprint of LRC".
>
> > Why does LRC outperform FP16 on so many cases?
>
>
> Thank you for your insightful remark. The evaluation of quantized models' accuracy on this type of benchmarks can be subject to noise, as quantized models may correctly answer some queries that full precision models fail to address. This phenomenon has been investigated in a recent study [1], where the authors suggest that distance metrics such as KL divergence should be used instead. In our work, we report both Perplexity (PPL), which is closely related to the KL divergence, and the accuracies obtained on various benchmarks to mitigate the noise.
>
> We hope that these clarifications adequately address your final concerns.
>
> Thanks again for reading our rebuttal, and for your detailed review.
>
>
> [1] Abhinav Dutta, Sanjeev Krishnan, Nipun Kwatra, Ramachandran Ramjee. "Accuracy is Not All You Need" (2024)

---

### Author Response · Authors · 2024-11-22
**Author Rebuttal by Authors**

**We thank the reviewers, AC and SAC assigned to this paper for their time and work looking into our submission.**

We thank them in advance for reading our rebuttal and interacting with us for a few more days during the discussion period.

We were happy to see that the paper was overall well received by all 5 reviewers:

**TKaQ:**  *They paper is written well and easy to understand. The scheme they propose is sound and intuitive in its formulation.*

**iJC7:** *The idea of introducing low-rank adaptation to correct the quantization error is good, and trivially effective.*

**RmtB:** *This innovative technique sets LRC apart from previous approaches and demonstrates its potential for enabling highly compressed models.*

**xtpe:** *They achieve a new SoTA for W4A4.*

**LQqC:** *The studied problem is important. The experimental results [...] show improvements over current methods, especially at W4A4 quantization levels.*

The most important weaknesses highlighted by reviewers point to:

- the lack of experimental results on the efficiency of our approach in term of memory and speed.

&#8594; *We have run novel experiments following their remarks. More precisely, we have implemented a cuda kernel using Cutlass to show the effect of the rank on the latency of LRC. Additionally, we have shared a table comparing the sizes in GB of the different LLMs considered in this work (unquantized, quantized without additional low-rank terms, and quantized with additional low-rank weight matrices).*

-  some clarifications on the effect of the calibration datasets.

&#8594; *We followed the suggestion of the reviewers and added an experiment to compare the performances obtained by Phi-3 when quantized at W4A4 using either wikitext2 or alpaca as the calibration dataset.*

- the missing proof.

&#8594; *We apologize for this omission. We have added all the proofs in the Appendix.*

We believe we have addressed all the points raised by the reviewers and have already implemented all the changes in our new version of the manuscript.

At the moment, all reviewers except **xtpe** have scored our paper as good (3) in presentation. We have received an average 2.4 and 2.2 grades in soundness and contribution respectively, and we hope our rebuttal alleviates these concerns.

We believe that the very supportive words found in all reviews are not reflected in the current distribution of (fairly low) scores of 5,5,5,5,3. If reviewers agree with our assessment, we humbly ask them to reconsider their score.

---

> ### Author Response · Authors · 2024-11-22
> **Experiment: Effect of the Calibration Dataset**
>
> We conducted a new experiment to investigate the impact of the calibration dataset selection on the performance of LRC. Our observations indicate that **the choice of the calibration dataset does not significantly affect the performance of the quantized models on downstream tasks**. Below, we present a comparison of LRC performance with a rank set to 10% of the original size on Phi-3 at W4A4. These results have also been added in Appendix B.1.
>
> Results with groupsizing (128)
> | Dataset   | Avg.  | A-c   | A-e   | HS    | LA    | PQ    | WG    |
> |-----------|-------|-------|-------|-------|-------|-------|-------|
> | Alpaca    | 0.7024| 0.5478| 0.7795| 0.7234| 0.6553| 0.7884| 0.7198|
> | wikitext2 | 0.7   | 0.5452| 0.779 | 0.7264| 0.6505| 0.784 | 0.7151|
>
> Results without groupsizing
> | Dataset   | Avg.  | A-c   | A-e   | HS    | LA    | PQ    | WG    |
> |-----------|-------|-------|-------|-------|-------|-------|-------|
> | Alpaca    | 0.6891| 0.5273| 0.7626| 0.699 | 0.6588| 0.7737| 0.7135|
> | Wikitext2 | 0.6917| 0.5341| 0.7782| 0.713 | 0.6511| 0.7835| 0.6906|

---

> ### Author Response · Authors · 2024-11-22
> **Experiment: Memory Footprint of LRC**
>
> Below, we present the sizes of various models (in GB) at W4. It is noteworthy that **low-rank methods incur approximately 13% additional weights relative to the original models when compared to QuaRot**. We have also added a column in Table 3 of the manuscript to report these memory footprints.
>
> | Method         | `FP16` | `QuaRot` | `SVD` | `LRC` |
> |----------------|--------|----------|-------|-----------|
> | **Phi**        | $6.75$ | $1.69$   | $2.59$| $2.59$    |
> | **Llama**      | $13$   | $3.25$   | $4.95$| $4.95$    |
> | **Mixtral**    | $86.5$ | $21.6$   | $32.1$| $32.1$    |

---

> ### Author Response · Authors · 2024-11-22
> **Experiment: Latency of LRC**
>
> Several reviewers inquired about the overhead associated with incorporating low-rank passes in FP16 with the int4 matrix multiplication. We did not address this in the paper for two main reasons: (a) numerous factors can influence timing, such as model size, batch size, hardware, and implementation specifics; and (b) developing an optimized kernel necessitates CUDA development, which is beyond our immediate expertise and outside the scope of this paper.
>
> Nonetheless, we provide here two artefacts that we hope the reviewers find useful: a discussion of the computational implications of the ranks, and a pytorch experiment with latency measurements.  These artefacts can also be found in Appendix B.2.
>
> Quantizing models is appealing because of both memory and latency improvements. In our experiments, we settled on 10% additional ranks because this seems like a fair trade-off in terms of memory: effectively we are at 5.6 bits (4 + 0.1 * 16). We argue that this additional footprint is worth spending for improved downstream-task performance.
>
> The additional FLOPS required are just 13% of the original model. Roughly speaking, int4 matmuls are twice as fast as fp16 on cuda devices, making a 'ballpark' estimate of our throughput 63% of FP16. But FLOPS are misleading: throughput on large models is often bottlenecked by data movement and deployment of models is often limited by footprint. As LLMs are deployed on specialist devices like Apple's silicon, or Qualcom/AMD accelerators on PC, the 'shape' of the hardware will lead to different latency results. In many cases, a mixed precision computation may be runnable using complementary parts of the hardware (e.g. int4 on an accelerator and fp16 on cpu).
>
> We set up this simple timing experiment on an Nvidia A100 device to time the cost of a forward pass. We use a batch size of 32, sequence length of 2048, and matrix sizes from the Llama model series. We used Cutlass to implement a basic int4 kernel. We timed the cost of quantizing the activations, computing the int4 kernel, computing the low-rank matmul in fp16, and adding the results. Our pytorch module looks like this:
>
> ```python=
> baseline_mod = torch.nn.Linear(feature_dim_in, feature_dim_out, bias=False).cuda().to(torch.float16)
> class Int4Lowrank(torch.nn.Module):
>     def __init__(self):
>         super().__init__()
>         self.quant = Quantizer(input_clip_ratio=1.0)
>         self.U = torch.nn.Linear(feature_dim_in, ranks, bias=False).to(torch.float16)
>         self.V = torch.nn.Linear(ranks, feature_dim_out, bias=False).to(torch.float16)
>         self.lin_4bit = Linear4bit.from_float(baseline_mod, weight_scales=s_w)
>     @torch.compile()
>     def forward(self, x):
>         return self.lin_4bit(self.quant(x)) + self.V(self.U(x))
> ```
>
>
> Here are the timings of this simple layer, with warmup, repeated 100x. Matrix sizes taken from the Llama family.
> | ranks    | matrix dim   | time (ms)       | speedup over fp16
> | -------- | --------     | --------        | ----
> | 0        | 11008x4096   | 13.89 +- 0.23   | 1.97
> | 128      | 11008x4096   | 18.04 +- 0.16   | 1.52
> | 256      | 11008x4096   | 19.019 +- 0.21  | 1.45
> | **512**      | 11008x4096   | 21.284 +- 0.2   | **1.29**
> | 1024     | 11008x4096   | 25.87 +- 0.26  | 1.06
>
> | ranks    | matrix dim   | time (ms)       | speedup over fp16
> | -------- | --------     | --------        | ----
> | 0        | 13824x5120   | 20.15 +- 0.03   | 2.03
> | 128      | 13824x5120   | 25.15 +- 0.09   | 1.63
> | 256      | 13824x5120   | 26.25 +- 0.05   | 1.56
> | **512**      | 13824x5120   | 29.140 +- 0.08   | **1.40**
> | 1024     | 13824x5120   | 34.77 +- 0.15   | 1.18
>
> | ranks    | matrix dim   | time (ms)       | speedup over fp16
> | -------- | --------     | --------        | ----
> | 0        | 28672x8192   | 54.83 +- 0.71   | 2.44
> | 128      | 28672x8192   | 64.40 +- 0.17   | 2.07
> | 256      | 28672x8192   | 66.77 +- 0.18   | 2.0
> | 512      | 28672x8192   | 72.03 +- 0.2    | 1.86
> | **1024**     | 28672x8192   | 82.98 +- 0.40    | **1.62**
>
> We see that **adding low-rank weight matrices does increase the latency of these operations as expected, though we still retain speedup relative to full FP16**. In each row of each table, we have highlighted the choice of ranks that is above (next power 2) the 10% factor we used in the main experiments in the paper.
>
> We have included numbers from very small ranks to emphasize a limitation of this experiment: even with a very small number of ranks added (128) there is latency loss. This implies that data movement is important, and that a fused kernel could improve latency.
>
> This experiment is also limited in that it does not account from groupsizing, which would make the addition of low-rank matrices _more_ appealing in terms of latency since int4 operations would themselves be reduced in speed.

---

### Meta-Review · Area_Chair_24eK · 2024-12-20

**Metareview:**

Dear Authors,

Thank you for your valuable contribution to ICLR and the ML community. Your submitted paper has undergone a rigorous review process, and I have carefully read and considered the feedback provided by the reviewers.

This work proposes a low-rank approach to enhance the performance of post-training quantization of large language models. The approach is evaluated on some recent language models.

The paper received mixed review scores (6,6,5,5,3). Reviewers pointed out critical issues including (i) over-simplified theoretical analysis, (ii) lack of a time complexity analysis, (iii) limited novelty of the method -- considering similar low-rank correction ideas recently proposed in the LLM quantization literature. Thank you for providing a detailed rebuttal. However, the rebuttal was not convincing enough for three reviewers to increase their scores.

Given the current form of the paper and the reviewer discussion, I regret to inform you that I am unable to recommend the acceptance of the paper for publication at ICLR. I want to emphasize that this decision should not be viewed as a discouragement. In fact, the reviewers and I believe that your work has valuable insights and, with further development and refinement, can make a meaningful impact on the field.

I encourage you to carefully address the feedback provided by the reviewers and consider resubmitting the paper. Please use the comments and suggestions in the reviews to improve and refine your work.

Best,
AC

**Additional Comments On Reviewer Discussion:**

Reviewers LQqC, RmtB and xtpe pointed out critical issues including (i) over-simplified theoretical analysis, (ii) lack of a time complexity analysis, (iii) limited novelty of the method -- considering similar low-rank correction ideas recently proposed in the LLM quantization literature. The authors provided a detailed rebuttal, however, the rebuttal was not convincing enough for three reviewers to increase their scores.

---

### Decision · Program_Chairs · 2025-01-22

Reject